# A highly dynamic mononuclear non-heme iron enzyme for the two-step isonitrile biosynthesis

Naike Ye [1], Antonio Del Rio Flores [2,3,4], Wenjun Zhang [4] ✉ & Catherine L. Drennan [1,5,6] ✉

The recent discovery of the isonitrile biosynthetic enzyme ScoE expanded the catalytic repertoire of the Fe(II)/αKG-dependent dioxygenase enzyme family. ScoE synthesizes an isonitrile functional group from a glycyl-fatty acid adduct, with both the isonitrile nitrogen and carbon atoms coming from the glycyl moiety. This challenging chemistry cannot be performed in a single step. Instead, the mechanism appears to require two half reactions, each involving αKG cleavage to generate a highly reactive iron-oxygen species. Here, we report sixteen crystal structures that provide snapshots along the reaction trajectory of Rv0097, a ScoE homolog from *Mycobacterium tuberculosis*. These structures, which are both of wild-type and Rv0097 variants, include a substrate 3-((carboxymethyl)amino)decanoic acid (CADA)-bound structure, an αKG-bound structure, and a structure with both CADA and αKG bound. These structural data reveal how Rv0097 employs conformational rearrangements to protect the unstable CADA-reaction intermediate that is formed in the first half reaction while swapping out αKG cleavage products for a second molecule of αKG. Additionally, these structures, together with data from site-directed mutagenesis, provide insight into Rv0097's preference for substrates with long alkyl chains, potentially facilitating efforts to re-engineer ScoE/Rv0097 to synthesize isonitrile functional groups on a wider range of small molecules.

The isonitrile functional group, with its zwitterionic character, is the subject of diverse chemical reactivities in organic syntheses. It acts as a nucleophile, an electrophile, a carbene, a radical acceptor, and a metal-binding ligand[1–4]. In nature, isonitrile can be found as a bioactive warhead. Since the initial discovery and identification of xanthocillin in 1950[5], hundreds of isonitrile-containing natural products have been isolated that participate in many biological processes, such as virulence, metal acquisition, and detoxification[6–8]. To date, however, only two isonitrile biosynthetic routes are known. The first route involves enzymes from the isonitrile synthase (IsnA) family that synthesize

isonitrile by condensation of the α-nitrogen atom from L-Tyr/L-Trp and a carbon atom from ribulose-5-phosphate (Supplementary Fig. 1A)[9]. The second route was reported recently in a study of conserved actinobacteria gene clusters involved in the biosynthesis of isonitrile lipopeptides (INLP)[10]. In the INLP biosynthetic gene cluster *ScoA-E* found in *Streptomyces coeruleorubidus*, ScoE is a non-heme Fe(II)/α-ketoglutarate (αKG)-dependent enzyme that catalyzes isonitrile formation by converting the glycyl moiety of a single substrate, (*R*)−3-((carboxymethyl)amino)butanoic acid (CABA), to an isonitrile group in the product, (*R*)−3-isocyanobutanoic acid (INBA) (Fig. 1A)[11]. In

[1]Department of Chemistry, Massachusetts Institute of Technology, Cambridge, MA, USA. [2]Department of Chemistry, Stanford University, Stanford, CA, USA. [3]Department of Chemical and Biological Engineering, University of Colorado, Boulder, CO, USA. [4]Department of Chemical and Biomolecular Engineering, University of California Berkeley, Berkeley, CA, USA. [5]Department of Biology, Massachusetts Institute of Technology, Cambridge, MA, USA. [6]Howard Hughes Medical Institute, Massachusetts Institute of Technology, Cambridge, MA, USA. ✉e-mail: wjzhang@berkeley.edu; cdrennan@mit.edu

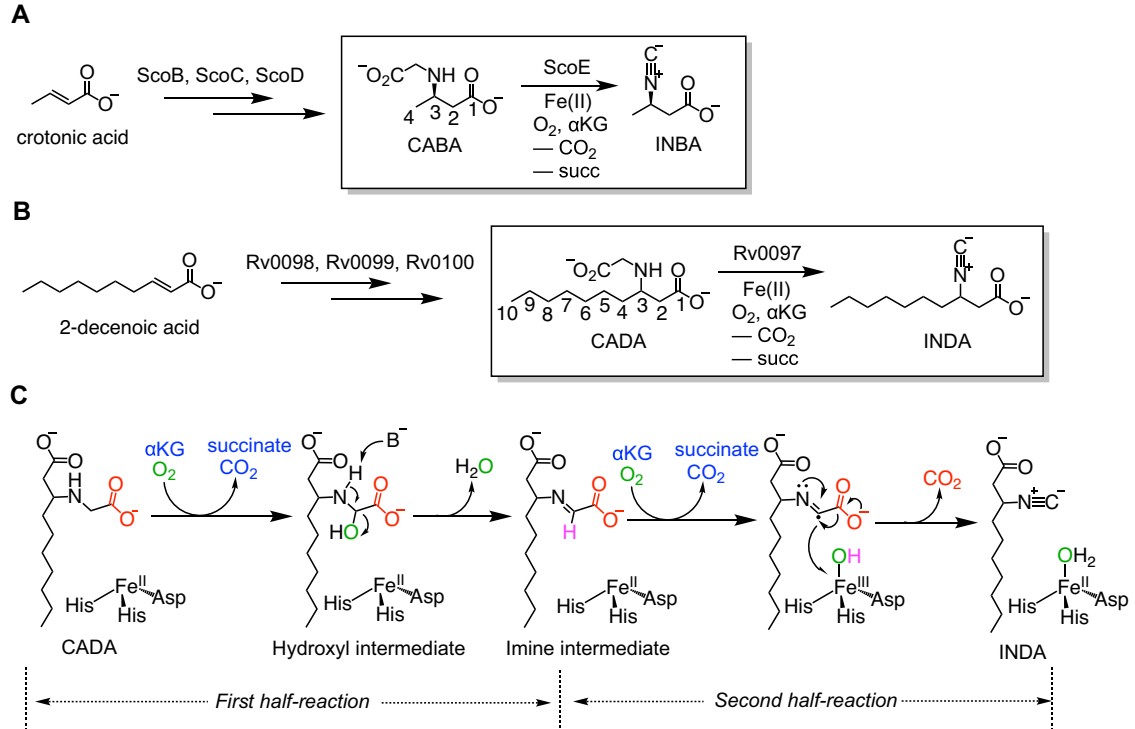

**Fig. 1 | ScoE and Rv0097 catalyze isonitrile biosynthesis. A** Starting with crotonic acid, a three-enzyme cascade synthesizes CABA, which is converted to INBA by the non-heme mononuclear Fe(II)/αKG-dependent dioxygenase ScoE. **B** Rv0097 is a ScoE homolog that converts CADA, synthesized from a 2-decenoic acid precursor, to INDA. **C** Rv0097 is proposed to perform isonitrile biosynthesis by an unusual two half-reaction mechanism.

a homologous gene cluster found in *Mycobacterium tuberculosis* H37Rv (*Mtb*), Rv0097 performs isonitrile biosynthesis on the glycyl moiety of substrates with long alkyl chains[12], including one with a 10-carbon fatty-acyl chain (C10), 3-((carboxymethyl)amino)decanoic acid (CADA), that is transformed to 3-isocyanodecanoic acid (INDA) (Fig. 1B)[12]. These ScoE/Rv0097-catalyzed isonitrile syntheses expanded the chemical repertoire of enzymes in the Fe(II)/αKG-dioxygenase superfamily.

Recently, we and others have characterized the ScoE-catalyzed reaction using both biochemical and structural approaches[13–16]. As a member of the non-heme Fe(II)-αKG-dependent dioxygenase enzyme family, ScoE relies on the cleavage of an Fe(II)-bound molecule of αKG by molecular oxygen to form a highly reactive iron-oxygen species, characterized in several family members to be an Fe(IV)-oxo intermediate[17–20]. This highly reactive Fe(IV)-oxo species can initiate a radical-based reaction by hydrogen atom abstraction from the substrate. For ScoE, we established the reaction stoichiometry and showed that two equivalents of αKG and molecular oxygen are required for every equivalent of CABA consumed and INBA produced[13]. Cleavage products of αKG: succinate and $CO_2$ are also produced[13]. We later reported putative intermediates and shunt products detected from the ScoE reaction and proposed a two half-reaction mechanism for isonitrile formation (Supplementary Fig. 1B)[14]. In this mechanism, the first half-reaction desaturates CABA using one equivalent of αKG and $O_2$ to form an imine intermediate (imine-CABA), which is then decarboxylated to form the isonitrile product by the second half-reaction using the second equivalents of αKG and $O_2$[14]. Although synthesizing an isonitrile group from a glycyl moiety is challenging, nature appears to overcome this hurdle through two sequential oxidation reactions mediated by two Fe(IV)-oxo intermediates. ScoE and its homologs are not alone in catalyzing multiple oxidative events in one catalytic cycle. Multifunctional Fe(II)/αKG-dependent dioxygenases are reported in the biosynthesis of fungal natural products that involve up to four rounds of oxidation reactions

(Supplementary Fig. 2)[21–24]. We are therefore interested in uncovering nature's design principles that allow for multiple reactions on a single substrate to afford chemically challenging conversions.

To accomplish two half-reactions in one catalytic cycle, ScoE faces several challenges in its enzyme design. First, αKG and CABA are highly structurally similar, thus the enzyme active site would need to differentiate between these two co-substrates. Second, the αKG and CABA binding sites must be close to each other for catalysis but have separate routes for entry and exit. Separate routes allow the enzyme to hold on to the hydrolysis-prone imine-CABA intermediate while exchanging succinate and $CO_2$ for a second molecule of αKG. Our previous ScoE structural studies[13] led to the proposal of two conformational changes for the binding of co-substrates (Supplementary Fig. 3); a small conformational change of K193 swinging into the active site to bind CABA, and a large (~10 Å) conformational change of R157/H299 loops flipping toward and away from the active site to allow the binding of αKG molecules by R157/H299 and release of reaction products. Because no structure of ScoE has been obtained that shows αKG bound in a catalytically-relevant fashion, the latter proposal is based on a tartrate-bound ScoE structure in which R157 was observed in the active site interacting with tartate[15]. Collectively, structural data from our lab[13] and others[15] started to unravel how ScoE performs isonitrile synthesis by the two half-reaction mechanism, but we are missing key pieces of the puzzle, including a structure with αKG bound in a catalytically competent position.

In this work, we shift our focus to Rv0097, an enzyme homologous to ScoE (45% identity, 61% similarity) from the gene cluster *Rv0097-Rv0101* found in *Mtb*. This gene cluster biosynthesizes an INLP involved in metal transport in *Mtb*, although the identity of the metal is still in debate[8,25,26]. Genetic studies showed that the *Rv0097-Rv0101* gene cluster is required for *Mtb* survival in mouse macrophages[27], and perturbations of the *rv0097* gene led to attenuated *Mtb* growth and survival in mice[28]. Therefore, a detailed mechanistic understanding of Rv0097 could contribute to structure-guided and mechanism-based

antibiotics design for tuberculosis treatments. Our biochemical studies suggested that Rv0097 recognizes substrates with a long alkyl chain (C8-C16) with a preference of >C10, whereas ScoE recognizes short-chain substrates (C4-C8) with a preference of C4[12]. Since the substrates of ScoE and Rv0097 only differ in alkyl chain lengths, we propose the same two half-reaction mechanism applies for Rv0097 (Fig. 1C, Supplementary Fig. 3).

Here, we add to the structural data on ScoE[11,13,15,16] and Rv0097[26] by obtaining a series of atomic-resolution crystal structures of Rv0097 in different states that enabled us to uncover the enzyme design principles that allow for the control of substrate specificity and allow for two oxidative events in one catalytic cycle. Based on our structural data, we propose a multi-layered conformational gating mechanism for the binding and exit of co-substrates and co-products that involve multiple sites of the enzyme. Our proposed mechanism is further supported by structure-guided mutagenesis and structural and biochemical studies on the variants.

## Results

We started by determining the 1.28-Å resolution crystal structure of Rv0097 apoprotein (apo) (structure **1** in Supplementary Table 1-2) by molecular replacement (MR) using our ScoE structure (PDB: 6XOJ)[13] (Fig. 2A). Rv0097 and ScoE share high structural similarity (all atom RMSD = 2.159 Å, Cα RMSD = 1.987 Å), both sharing the same double-stranded β-helix fold, also known as the jellyroll or cupin fold, which is common among enzymes in the family of Fe(II)/αKG-dependent dioxygenases[29]. The asymmetric unit (ASU) contains a homodimer that is held together by extensive hydrophobic interactions and a helix-loop region (Supplementary Fig. 4). Next to the dimer interface region are the substrate-binding site and Fe site (Fig. 2A). A conformationally flexible region (Fig. 2A) contains conserved His and Arg residues (R157/H299 in ScoE; R122/H264 in Rv0097). In the Fe site, we observe electron density for a metal that is ligated to the His$_2$Asp$_1$ triad, two water molecules, and an acetate ion (Fig. 2B). Since an ethylenediaminetetraacetic acid (EDTA) chelation step was included in protein purification to prevent adventitious binding of metals (see Methods) and X-ray fluorescence of the resulting protein crystal indicated successful transition metal chelation, we modeled a Mg$^{2+}$ ion from the crystallization solution into this density (Fig. 2B). Alternative modeling of a water molecule into this density results in positive difference density (Supplementary Fig. 5) and modeling of heavy metals (such as Fe or Zn) results in negative difference density (Supplementary Fig. 5). In the absence of CADA, an acetate ion is bound in the substrate-binding site. This acetate hydrogen bonds to Y70 and K158 sidechains, and to a chain of water molecules that lead to the Mg$^{2+}$ ion (Fig. 2B).

### A pair of residues in the substrate-binding pocket gates substrate binding and specificity

Leading to the active site in the putative substrate-binding pocket that is occupied by acetate in structure **1**, we observe two rotamer conformations of the F102 residue (F102 and F102′) (Fig. 2C). Structure refinements with only one of the two F102 conformations led to significant negative difference density at the modeled rotamer and positive difference density at the unmodeled rotamer, suggesting the presence of both conformations at partial occupancies (Fig. 2C). An omit map also shows the density for both F102 conformations (Fig. 2C). Refined occupancy suggests both rotamers are present roughly half of the time (0.48/0.52). Adjacent to this F102 residue is G204, a residue that was previously identified to be non-conserved across homologs (Gly in Rv0097 and MmaE, but Phe in ScoE (F239))[12]. Structural overlay of Rv0097 with ScoE (PDB: 6XOJ) shows steric clashes between the F102′ rotamer of Rv0097 and F239 of ScoE, indicating that the alternate conformation of F102 is enabled by the lack of a side chain for G204 as opposed to its counterpart residue F239 in ScoE (Fig. 2D).

To examine the role of this F102/G204 residue pair in substrate binding in Rv0097, we obtained a 1.12-Å resolution crystal structure with Fe(II) and CADA co-crystallized (Fig. 2E, structure **2** in Supplementary Table 1-2). We note here that we used racemic-CADA for co-crystallization due to substantial costs and time investments it requires to synthesize the enantiopure (R)-CADA[26]. We find only the (R) isomer bound in the structure, indicating the preference of the enzyme for (R)-CADA over (S)-CADA, as has been previously reported[26]. This Fe(II)-CADA bound structure is highly similar to the previously reported Rv0097 structure with Fe(II) and CADA bound[26] (PDB: 8KHT) (all atom RMSD = 1.126 Å, Cα RMSD = 0.613 Å). Since CADA has a polar moiety that is identical to the polar moiety of ScoE's substrate CABA and a nonpolar alkyl chain that is longer (seven carbon units) than that the single carbon unit (methyl) moiety in CABA, Rv0097 has a binding site that is both polar and nonpolar (Fig. 2E). The conformation and interactions of the polar part of CADA in Rv0097 are very similar to the conformation and interactions of CABA observed in ScoE (Supplementary Fig. 6). The two negatively charged carboxylates of CADA form charge-charge interactions with side chains of K158 and R275 (Fig. 2E). One carboxylate additionally forms a hydrogen bond with Y70. The Y65 side chain hydrogen bonds to the nitrogen of the glycine moiety of CADA. The long alkyl chain of CADA sits snugly in the substrate-binding site, stabilized by hydrophobic interactions with F95, M101, F102, P153, H156, C201, T203, and G204 (Fig. 2E). These substrate-binding pocket residues are conserved in Rv0097 and MmaE, but not in ScoE (Supplementary Fig. 7). With CADA bound, the F102 residue is now only in one conformation (F102), creating space to accommodate CADA's long alkyl chain (Fig. 2E, right, Supplementary Fig. 8). Differential positioning of two loops (pocket loops 1 and 2, Supplementary Fig. 9) in Rv0097 and ScoE additionally create more space for the long alkyl chain of CADA. These structural observations suggest that substitutions of G204 to residues with bulky side chains, such as Phe, would inevitably lead to substantial steric clashes in the presence of the CADA, consistent with a previous report showing that G204F variant dramatically compromises enzymatic activity (as measured by product formation) and perturbs substrate specificity of Rv0097[12]. To examine the structural effect of a G204 substitution, we obtained crystallographic snapshots of a G204F variant without substrate (structure **3** in Supplementary Table 1-2), in complex with the longer CADA substrate (structure **4** in Supplementary Table 1-2), and in complex with the shorter CABA substrate (structure **5** in Supplementary Table 1-2). In the absence of substrate (structure **3**), the distance between the G204F and F102 side chains is ~3.8 Å, too close to allow CADA's 10-carbon alkyl chain to bind in between the two residues (Fig. 2F). Overlay of structures **3** and **2** shows steric clashes between G204F side chain and C8, C9, C10 of the CADA substrate molecule (Fig. 2G), supporting our previous data showing ScoE's strong preference for short-chain alkyl chain substrates (C4) and Rv0097's strong preference for long-chain alkyl substrates ( > C10)[12].

Consistently, structure **4** shows weak electron density for CADA in the presence of the G204F substitution (Fig. 2H). Density is present for the polar end of the substrate, but not for the alkyl chain at its typical binding site. Instead, curved density above G204F is observed in the omit map into which we modeled the alkyl chain of the CADA (Fig. 2H). This curled-up binding pose of CADA is likely to have a considerable steric strain associated with it, explaining why CADA is not present at full occupancy in the structure. The occupancy refined to 0.54 when the B-factors of CADA were set to match the B-factors of neighboring residues. Similarly, structure **5** shows poor electron density of the shorter-chain CABA, consistent with its weak binding to the active site, with refined occupancy of 0.65 (Fig. 2I).

We additionally obtained kinetics data of the G204F variant for both CADA and CABA (Supplementary Fig. 10). As with the structural studies, we used a racemic mixture of CADA in all our assays. Since we don't see evidence of (S)-CADA binding in our structures, we believe

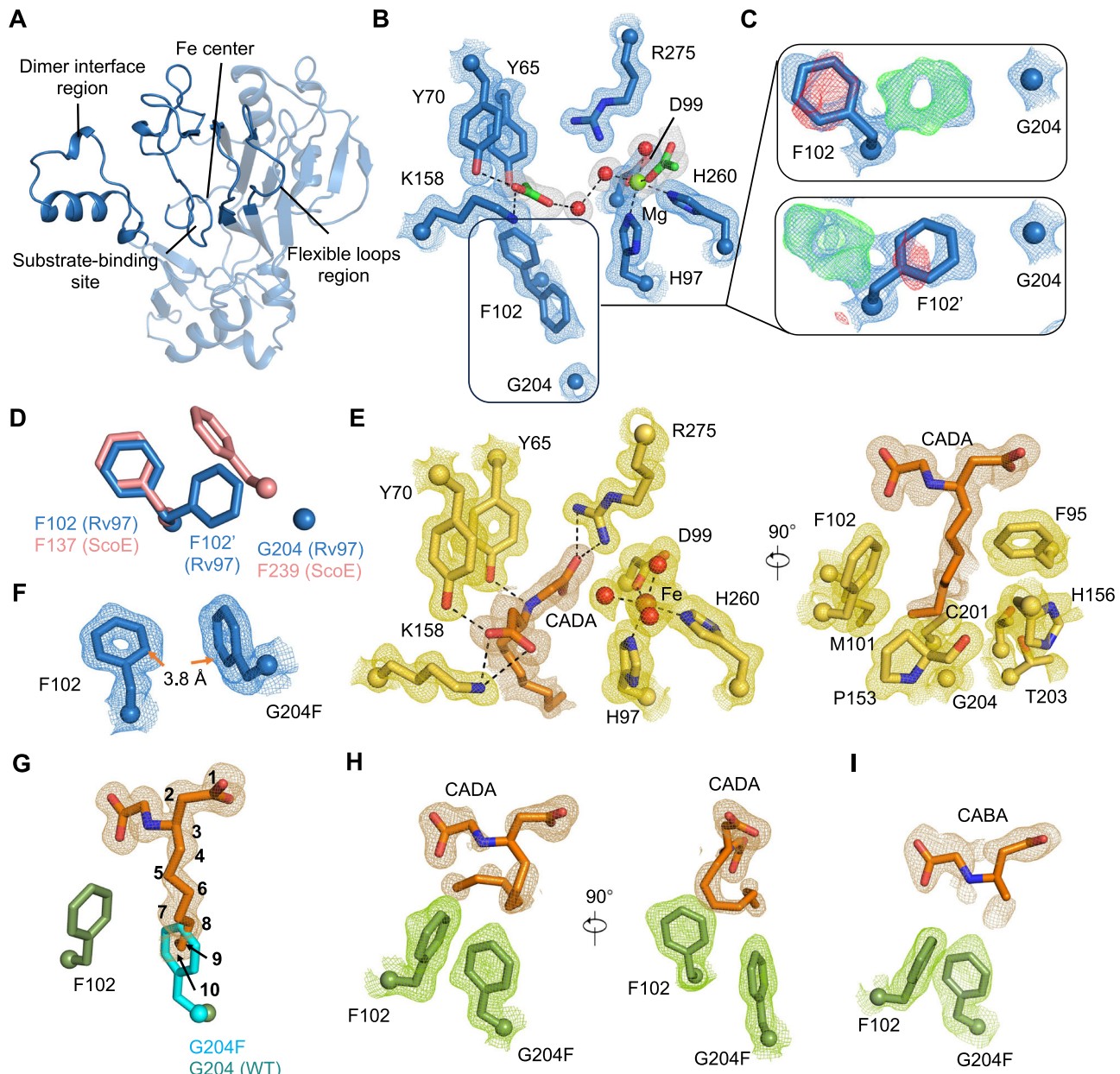

**Fig. 2 | F102/G204 residue pair in Rv0097 governs substrate binding and specificity. A** Overall architecture of Rv0097. **B** Active site of the apo structure (structure **1**) shown in a 1.0 σ 2mF$_o$-DF$_c$ composite omit map. One acetate ion is bound in the substrate-binding site and another one is bound to Mg$^{2+}$. Water molecules are shown as red spheres. The two conformations of F102 (occupancies 0.48 and 0.52) and its neighboring residue G204 are boxed. **C** Modeling of a single F102 rotamer results in mF$_o$-DF$_c$ difference density (+3.0 σ in green mesh, −3.0 σ in red mesh). **D** Overlay of F102/G204 and their counterparts in ScoE show that the non-conserved G204 residue enables alternate conformations of F102. **E** Active site of the Fe ion and CADA co-crystal Rv0097 structure (structure **2**) shown in a 1.0 σ

2mF$_o$-DF$_c$ composite omit map. To bind CADA, which has a polar part and also a nonpolar alkyl chain, Rv0097 has an active site that is both polar (left) and nonpolar (right). Black dashed lines represent hydrogen bonds or ligations to the metal. **F** The G204F variant apo structure (structure **3**) shows close contact between F102 and G204F. **G** Overlay of structures **2** and **3** shows direct overlap of the alkyl chain of CADA and the G204F side chain. **H** Vanadyl-CADA-G204F structure (structure **4**) with "curled up" CADA bound in the substrate-binding site, with refined occupancy of 0.54. **I** Vanadyl-CABA-G204F structure (structure **5**) presents weak density for CABA. 2mF$_o$-DF$_c$ composite omit maps are contoured to 1.0 σ.

that the impact of the racemic CADA mixture on enzyme activity is minor, consistent a with previous report[26]. However, the use of a racemic mixture of CADA and enantiomeric pure CABA is a caveat that should be kept in mind. We find that the enzyme kinetic parameters ($k_{cat}$ and $K_M$) and the catalytic efficiencies ($k_{cat}/K_M$) are similar for both substrates with the G204F variant. The $k_{cat}$ values are similar between the G204F and wild-type enzyme (WT), indicating that when substrate is saturated, the G204F variant catalyzes isonitrile formation at the same rate as WT. However, the $K_M$ for the G204F variant against both substrates are much higher than that of WT (kinetics parameters for

WT enzyme are from our previous work[12]), indicating weakened binding affinity of G204F towards both substrates. Due to the higher $K_M$ values, the catalytic efficiencies of the G204F variant on both substrates are decreased compared to WT; less than or equal to 30% of the WT enzyme's catalytic efficiency on CADA (Supplementary Fig. 10).

To delve further into the roles of the two conformations of F102 in gating substrate binding, we prepared three F102 variants: F102W, F102Y, and F102A. We obtained a series of structures of these variants in apo form, in complex with CADA and a vanadyl ion, an Fe(IV)-oxo mimic[30–32] (structures **6-10** in Supplementary Table 1-2), or in complex

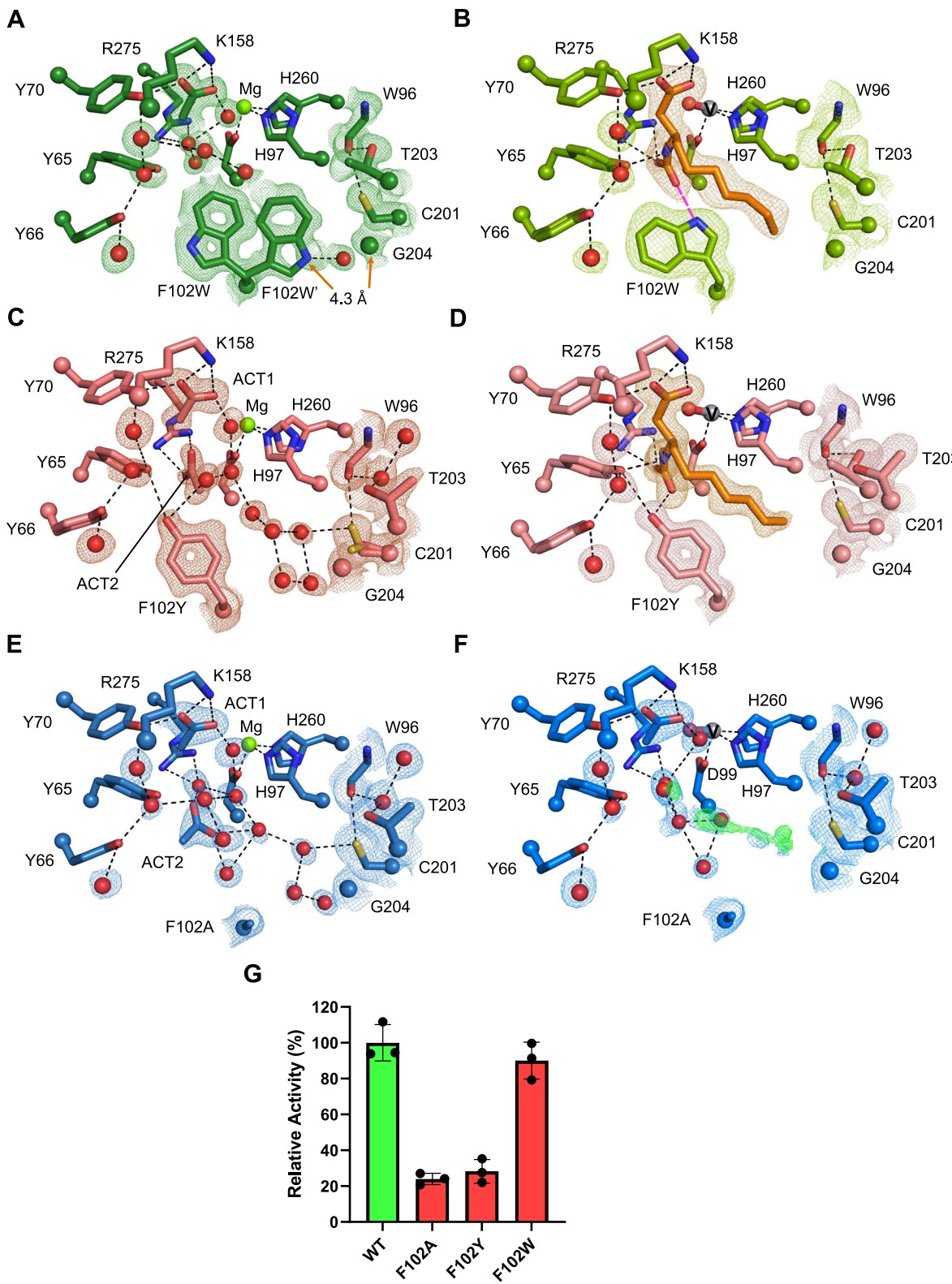

with a vanadyl ion alone (structures **11** in Supplementary Table 1-2). First, we changed F102 to a similar but bulkier aromatic residue Trp (F102W). Notably, in its apo structure at 1.25-Å resolution (structure **6** in Supplementary Table 1-2), the residue W102 is also present in two rotamer conformations (Fig. 3A). The F102W rotamer is more dominant (occupancy= 0.63) than the F102W' rotamer (occupancy= 0.37). An acetate ion and water molecules are found at the polar end of the CADA binding site. The non-polar end of the CADA binding pocket is

still hydrophobic, largely due to the two rotamer conformations of F102W, which seem to prevent water molecules from residing in the cavity. Only one water molecule can be found hydrogen bonded to the nitrogen of F102W's indole ring. Like the WT enzyme, the distance between the F102W' rotamer (nitrogen of indole ring) and G204 alpha carbon is short ( ~ 4.3 Å) (Fig. 3A). An additional entry point for solvent into the CADA alkyl chain binding pocket is found at residue T203. In structure **6**, this entry route is closed due to the side chain of T203

**Fig. 3 | Structures and assays of F102 variants. A** Structure **6** shown in a 1.0 σ 2mF$_o$-DF$_c$ composite omit map. Water molecules fill the polar part of the substrate-binding site. Two rotamer conformations of F102W keeps the alkyl chain binding site largely hydrophobic. Water molecules are shown as red spheres. Mg ions are shown as green spheres. Black dashed lines represent hydrogen bonds or ligations to metals. **B** Structure **7** in a 1.0 σ 2mF$_o$-DF$_c$ composite omit map. When CADA is bound (shown in orange sticks), F102W side chain is only in one conformation and forms an additional hydrogen bond with CADA carboxylate (shown in magenta dashed line). Vanadyl ion is shown in sticks with V in gray and the oxo in red. **C** Structure **8** in a 1.0 σ 2mF$_o$-DF$_c$ composite omit map. F102Y side chain is locked in one conformation. The substrate-binding site is occupied by a water molecule network that extends to the protein surface. **D** Structure **9** in a 1.0 σ 2mF$_o$-DF$_c$ composite omit map. CADA binding makes the substrate-binding site hydrophobic but T203 alternate conformations could still allow water access from the protein surface. **E** Structure **10** in a 1.0 σ 2mF$_o$-DF$_c$ composite omit map. An extended water network can be observed. **F** Structure **11** in a 1.0 σ 2mF$_o$-DF$_c$ composite omit map. Electron density for CADA cannot be clearly observed in mF$_o$-DF$_c$ difference map (+2.8 σ in green mesh). **G** Activity assays conducted with a 10-min incubation period and using a non-saturating concentration of CADA (1 mM) show relative product formations of three F102 variants in comparison to the WT enzyme. For each assay, data and error bars represent average and standard deviation from three independently performed experiments, respectively. Source data are provided as a Source Data file.

adopting a conformation that allows for a hydrogen bond to the backbone carbonyl carbon of W96 (Fig. 3A), which seals off access to bulk solvent. When CADA is bound, our 2.00-Å resolution structure (structure **7** in Supplementary Table 1-2) shows that only one of the two conformations of F102W is observed, like that in WT (Fig. 3B). However, the single conformation of F102W that is observed is rotated by 180° to avoid steric clashes with the carboxylate of the glycyl moiety of CADA (Fig. 3B). This rotation creates an additional hydrogen bond between the indole nitrogen of F102W and the carboxylate oxygen of CADA (Fig. 3B, magenta dashed line). Overall, when CADA is bound, the substrate-binding site is hydrophobic; water molecules cannot access the pocket at the F102 position or at the T203 position.

We then prepared the F102Y variant and determined its apo crystal structure to 1.37-Å resolution (structure **8** in Supplementary Table 1-2). With an added hydroxyl group in the side chain, F102Y is locked in one conformation by a new 3.2-Å hydrogen bond formed between the hydroxyl groups of Y102 and Y65 (Fig. 3C). Water molecules occupy the site left open by the lack of a second F102Y side chain conformer, filling with solvent molecules the otherwise hydrophobic binding pocket for the alkyl chain of substrate (Fig. 3C). This observation supports a role for F102 in gating access to the substrate-binding pocket. This structure (**8**) additionally shows that the solvent entry point at T203 is open. The side chain of T203 adopts a different rotamer conformation that creates an opening between the pocket and bulk solvent (Fig. 3C). This alternative T203 side chain conformation is stabilized by a hydrogen bond between the T203 side chain hydroxyl group and a water molecule, which in turn has a hydrogen bond with W96's backbone carbonyl oxygen (Fig. 3C). When excess CADA is added to the protein for co-crystallization, we find in the resulting 1.46-Å resolution structure (structure **9** in Supplementary Table 1-2) that CADA has displaced the water molecules and that T203 is now sampling both rotamer conformations at approximately equal occupancy (0.54 open to 0.46 closed) (Fig. 3D). Thus, CADA can still bind to the pocket in the F102Y variant, and that binding influences the position of T203.

For the F102A variant, the crystal structure of the apo form at 1.73-Å resolution (structure **10** in Supplementary Table 1-2) shows water molecules taking the place of the bulky side chains of F102 in WT and F102W/F102Y in the variants (Fig. 3E). T203 is in the rotamer conformation that generates the opening to bulk solvent (Fig. 3E). For this variant, even 5-fold excess CADA in the crystallization does not lead to substantial density for CADA's alkyl chain. The crystal structure at 1.82-Å resolution (structure **11** in Supplementary Table 1-2) shows the side chain of T203 in the open conformation, water molecules in the substrate-binding site, and no significant density for CADA (Fig. 3F). When the difference map is contoured to +2.8 σ, a continuous piece of positive density appears that is reminiscent of the alkyl chain of CADA. However, there is no corresponding density observable in the omit map, hence we did not model CADA into this active site.

To validate the importance of F102, we turned to enzyme activity assays (see Methods). First, both the F102W variant and the F102A variant share slightly lower $k_{cat}$ values compared to the WT enzyme

## Table 1 | Kinetic parameters for Rv0097 WT and the F102W and F102A variants for CADA

| Enzyme | $K_M$ | $k_{cat}$ (min$^{-1}$) | $k_{cat}/K_M$ (mM$^{-1}$min$^{-1}$) |
|---|---|---|---|
| WT | 405 ± 33 µM | 3.3 ± 0.4 | 8.1 ± 1.1 |
| F102W | 548 ± 80 µM | 3.2 ± 0.3 | 5.8 ± 1.0 |
| F102A | 1.2 ± 0.2 mM | 3.2 ± 0.3 | 2.5 ± 0.5 |

(Table 1, Supplementary Fig. 11), indicating that in substrate-saturating conditions, all three enzymes perform catalysis at comparable rates. However, these three enzymes differ in $K_M$ values. The F102W variant has a slightly higher $K_M$ value for CADA than WT whereas the F102A variant has a much higher $K_M$ value for CADA (in mM range) than WT. Due to these differential $K_M$ values, the catalytic efficiency ($k_{cat}/K_M$) of the F102A variant is less than one half of F102W and less than one third of WT, which is the most catalytically efficient (Table 1). In a product formation assay for WT and the three F102 variants, which was conducted with a 10-min incubation period using 1 mM CADA, F102W produced the same amount of product as WT within the error of the experiment, whereas F102A and F102Y variants showed substantially decreased product formation (Fig. 3G). All of these data are consistent with our structural observation that F102W's substrate-binding site is substantially more hydrophobic with far fewer water molecules for CADA to displace (Fig. 3A) than that of F102A (Fig. 3C) and F102Y (Fig. 3E), hence better suited for binding of the hydrophobic substrate, CADA[33,34].

## Rv0097 has both open and closed conformations that provide access to the active site

Our co-crystallization and soaking experiments with Fe(II) and/or Fe(IV)-oxo mimic vanadyl (aka oxovanadium(IV) cation)[30–32], with CADA and/or αKG, of the WT and variant protein have led to multiple crystal forms (Supplementary Table 1-2). These different crystal forms have allowed us to capture many conformational states of Rv0097. Some of the conformational rearrangements that we have observed are on the smaller side like the rearrangements of the side chains of T203 and of F102 mentioned above. Others are more dramatic and involve substantial loop movements and secondary structure changes (Fig. 4A-B). Fortunately, the Rv0097 crystal lattice turned out to be remarkably malleable, allowing us to obtain crystallographic snapshots of numerous conformational states. Although the soaking of our anaerobically grown Fe(II)-Rv0097 co-crystals in crystallization solutions supplemented with 5 mM CADA for 1 h caused crystals to crack (Supplementary Fig. 12), these damaged crystals still diffracted to high resolution and a 1.74-Å resolution structure was obtained (structure **12** in Supplementary Table 1-2). Indexing of data indicated the same space group ($P2_1$) but an expanded unit cell with four protomers in the ASU, instead of two, and a solvent content of ~42%. Close examination of packing in the crystal lattice reveals that the new ASU contains a homodimer like in previous structures (Supplementary Fig. 13A-C) and

two additional protomers each from a different homodimer (Supplementary Fig. 13D, E). Since the space group remains the same, this new ASU with four chains is related to another ASU along the *b* direction by a screw axis (Supplementary Fig. 13F).

Although CADA is not bound in any of the structure **12** protomers, this soaking experiment led to a structure with a wide-open active site (open state) and to three structures in which the active site is closed (closed state) (Fig. 4A). Overlayed structures of open (pink in Fig. 4A) and closed states (yellow in Fig. 4A) (all atom RMSD = 2.295 Å, Cα RMSD = 2.189 Å) reveal significant conformational rearrangements of the dimer interface region ("left lid", residues D152 to P181) and a region on top of the active site ("right lid", residues Y65 to F95) (Fig. 4A). The movement of the "right lid" affects the positioning of two surface loops that are important for αKG positioning; the His loop and Arg loop, which are discussed in the following section. The movement of the "left lid" involves a 18.6° rotation to open the active site (Fig. 4A, green angle). The first half of the "right lid", residues Y65 to I78, undergoes an upward ~2.5 Å shift (Fig. 4B, green arrow), whereas the second half, I78 to F95, undergo more drastic and complex conformational rearrangements that involve changes in protein secondary structures (Fig. 4B). In the closed state (yellow in Fig. 4), residues I78-S82 form a short β-strand (strand 1) that hydrogen bonds to an adjacent β-strand (strand 2), followed by a long-ordered loop (residues S82-F95). In the open state (pink in Fig. 4), residues S82-E85 undergo a loop-to-strand secondary structure change, and the tail of strand 1 is extended from S82 to E85. Strand 2 is also extended with residues G269-T272 undergoing a similar loop-to-strand rearrangement that results in strand 2's N-terminus starting at residue G269 instead of at T272. The extended parts of both strands are held together by backbone hydrogen bonds. Residues E85-G93 in the open state are disordered, thus exposing the active site to the bulk solvent (Fig. 4C). A pool of ordered water molecules occupies the CADA-binding site and the site near R122/H264 where αKG is expected to bind (Fig. 4C). Both R122 and H264 are pointing toward the putative αKG binding site even though αKG is not present (Fig. 4C). In contrast, the active site in the closed state has far fewer solvent molecules (Supplementary Fig. 14A). Our observed open and closed protein conformational states have been seen before in the family of mononuclear non-heme iron enzymes. In the prototypical non-heme Fe(II)/αKG-dependent dioxygenase TauD, substrate taurine binding induces a conformational change of TauD from open to closed[35] (Supplementary Fig. 15). Like Rv0097, a "left lid" and a "right lid" could be identified that undergo structural rearrangements as substrate binds in the active site (Supplementary Fig. 15). However, no secondary structure change is observed.

Since soaking of anaerobically grown Fe(II)-Rv0097 co-crystals with 5 mM CADA did not yield substrate density, we doubled the soaking concentration of CADA to 10 mM and solved the structure to 1.76-Å resolution (structure **13** in Supplementary Table 1-2). Again, this crystal form contained four chains in the ASU, one open state structure and three closed state structures. In all three of the closed state structures, clear density corresponding to CADA molecules, and not acetate molecules, were now present (Supplementary Fig. 14B, C). To obtain a structure with both CADA and αKG bound, we soaked anaerobically grown Fe(II)-Rv0097 co-crystals with 5 mM of both CADA and αKG. We obtained a 1.82-Å resolution structure (structure **14** in Supplementary Table 1-2) in which we observed strong electron density that corresponds to an αKG bound bidentate to Fe(II) (Fig. 4D). However, no density was observed for CADA. We find that αKG is bound to Rv0097 with its C5 carboxylate positioned to form a salt bridge with the side chain of R122 (Fig. 4D). Based on the ScoE structure with tartrate, we proposed that R122 of Rv0097 and the equivalent ScoE arginine (R157) would be responsible for anchoring αKG in the active site. Now we know that for Rv0097, our proposal is correct.

However, this αKG is bound to the Fe site in what appears to be an offline orientation with the Fe-binding C1 carboxylate oxygen

occupying the expected position of the putative Fe(IV)-oxo intermediate. αKG is said to be offline when its C1 carboxylate oxygen is *trans* to the second His in the His...Asp(Glu)...His triad, requiring that the Fe(IV)-oxo is *cis* to that second His[36] (Supplementary Fig. 16). With αKG bound in this manner, the only open coordination site on the Fe(II) ion is facing away from the substrate binding site, meaning that the Fe(IV)-oxo intermediate formed by the molecular oxygen-dependent cleavage of αKG would not be positioned correctly for substrate hydrogen atom abstraction. Thus, we expect that either αKG rearranges upon substrate or molecular oxygen binding or that the Fe(IV)-oxo intermediate flips closer to substrate following the αKG cleavage reaction. Both αKG rearrangements[37] and Fe(IV)-oxo intermediate flips[38] have been proposed in the mechanisms of other Fe(II)/αKG-dependent dioxygenases.

In a further attempt to obtain a Fe(II)•αKG•CADA•Rv0097 ternary complex structure, we co-crystallized Fe(II), CADA and Rv0097, and then soaked these crystals in 10 mM αKG. The resulting structure at 1.45-Å resolution (structure **15** in Supplementary Table 1-2) excitingly shows density for both CADA and αKG (Fig. 4E). We find both R122 and H264 pointing toward the Fe center and that αKG is bound to Fe(II) in the same putative offline orientation, as described above, again making the salt bridge interactions between C5 carboxylate of αKG and R122 side chain. CADA binds to the substrate-binding site with no steric clashes with αKG, indicating that this offline αKG conformation can accommodate substrate in the active site (Fig. 4E). Structure **15** suggests that it is not substrate binding that leads αKG to adopt an online conformation (see Supplementary Fig. 16). However, since the αKG was soaked rather than co-crystallized, it is possible that the enzyme was not able to undergo the conformational change required for an online αKG conformation, and that in solution, substrate binding does cause αKG to adopt an online conformation. Alternatively, as mentioned above, it is also possible that molecular oxygen binding to Fe(II) triggers a conformational change in αKG or that this conformation is in fact online, and that it is the Fe(IV)-oxo intermediate that rearranges. More work is needed to resolve these specific questions.

## Conformational changes associated with open and closed states affect CADA-gating residues

The conformational changes associated with the open and closed enzyme structural states directly affect substrate access to the active site and substrate-binding residues (Fig. 4F). Notably, in the open conformation, F102 adopts the F102′ conformation and point towards G204. Surface representation of the open chain suggests that the active site is accessible for substrate binding (Fig. 4G, circled region). F102′ directly stacks against a neighboring P153 residue and is stabilized by C-H(P153) ···π(F102′) interaction, with centroid-to-centroid distance of 4.1 Å (Fig. 4H). Such C-H···π interactions between an aromatic residue and a Pro residue has precedence in protein structures[39]. For example, in the glycyl radical enzyme HypD, the substrate *trans*-4-hydroxy-L-proline is stabilized by a proline-aromatic interaction with F340 residue[40]. Sequence and structural alignments of Rv0097 with ScoE indicate that this P153 residue is not conserved. In ScoE, V188 takes the place of P153, and is similarly stacked on top of F102's counterpart F137, with a distance of 3.7 Å (Supplementary Fig. 17). To test the importance of P153 in Rv0097, we prepared two enzyme variants of this P153 residue, P153V and P153A. Activity assay monitoring product formation on both variants showed decreased activity (Fig. 4I), suggesting that weakening this C-H···π interaction probably destabilized the open conformation of the enzyme, thereby affecting substrate binding, hence leading to lower enzymatic activity.

## Movements of a His loop and Arg loop provide a different route into the active site for second αKG

Our previous work on ScoE[13] showed that both H299 and R157 are important for catalysis. Here we show the same is true for Rv0097's

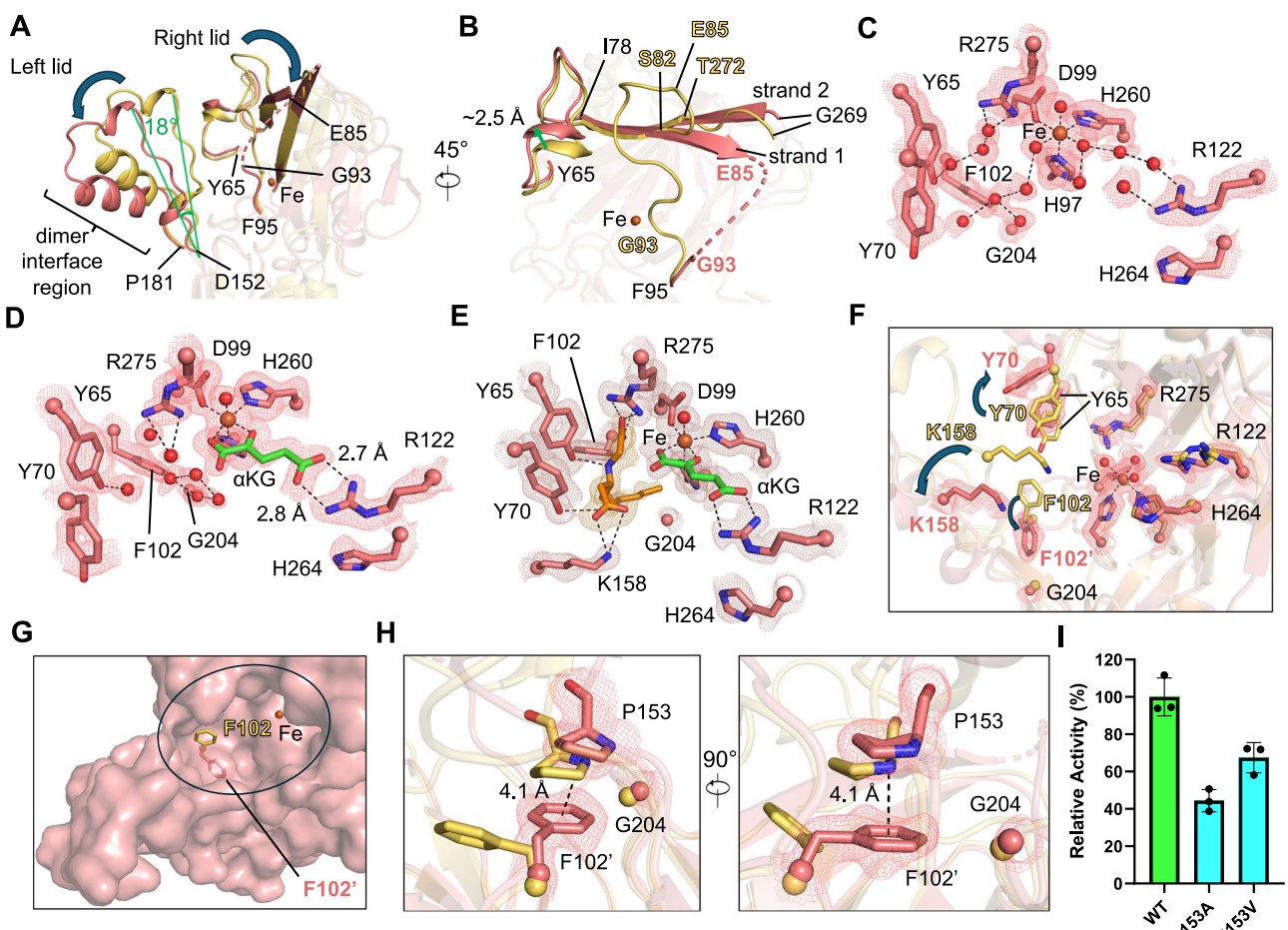

**Fig. 4 | Enzyme open conformation allows substrate and αKG binding.**
**A** Overlayed structures of **2** (yellow) and the open conformation (chain C) of **12** (pink). The angle marked with green lines indicates the degree of rotation of the "left lid" from closed to open states. Blue arrows indicate conformational changes. **B** Rotated view highlighting conformational changes of the "right lid" region. The green arrow indicates upward shift of the Y65-I78 part of the "right lid". **C** Active site of the open chain of structure **12** shown in a 1.0 σ 2mF$_o$-DF$_c$ composite omit map. The active site is occupied by water molecules that are shown as red spheres. Black dashed lines represent hydrogen bonds or ligations to metals. **D** Active site of the open state structure **14** (salmon). αKG (shown in green sticks) is bound to Fe in an offline orientation. **E** Active site of structure **15**. αKG is bound in the offline conformation. CADA is shown as orange sticks. **F** Zoomed-in view of the active site of overlayed structures of **2** and **12** revealing conformational changes of substrate-binding residues. Blue arrows indicate conformational changes of residues from closed to open conformations. **G** Surface representation of the open state structure from **12**. The black oval indicates the active site, which is directly solvent exposed. **H** P153 stacks against F102' and forms C-H···π interactions. The black dashed lines indicate centroid-to-centroid distance. **I** Activity assays showing relative product formations of P153 variants compared to WT. For each assay, data and error bars represent average and standard deviation from three independently performed experiments, respectively. Source data are provided as a Source Data file.

H264 and R122 (Fig. 5A). Activity assays monitoring product formation on H264/R122 variants show that both residues are critical for enzymatic activity (Fig. 5B), with R122 being the more important one, likely because of its essential role in αKG coordination. Based on the ScoE structure with tartrate bound, we proposed an inducible αKG-binding mechanism for ScoE in which loops containing H299 and R157 undergo large conformational changes to create a binding site for αKG every time that αKG binds. Arg is a common αKG-binding residue, but in other Fe(II)/αKG-dependent dioxygenases, the Arg does not move and the αKG binding site is not inducible[17]. We speculated a few reasons why ScoE might have evolved an inducible mechanism for αKG binding, including that such a mechanism would aid in binding specificity for two co-substrates (CABA and αKG) that are structurally similar to each other. Given that CADA is less similar to αKG due to CADA's long alkyl chain, CADA would be less likely to compete with αKG for binding to the same site, potentially negating the need for an inducible αKG-binding site. In this work, we find that Rv0097 does have an inducible αKG-binding site. We observe different conformations of the loops (the His loop and the Arg loop), which contain the analogous residues to ScoE's H299 and R157; Rv0097's H264 and R122 (Fig. 5A).

Overlaying the WT structures (**1, 2, 12, 15, 16**) and ScoE structures (PDB: 6XO3[13], 6L6X[15]) reveals different conformations for H264/H299 and R122/R157 side chains, which can be generally classified as inward (toward active site) and outward (away from active site) (Fig. 5C). Interestingly, a 1.36-Å resolution vanadyl and CADA bound structure (structure **16** in Supplementary Table 1-2) shows additional movement of H264 further towards the active site in a position analogous to that of H299 of ScoE with tartrate bound (Fig. 5C). Like ScoE, CADA binding to Rv007 did not promote a conformational change in the His/Arg residues: both apo (structure **1**, Supplementary Fig. 18) and the Fe(II)- and CADA-bound structure (structure **2**) have R122 and H264 residues in the outward conformation. However, when Rv0097 is in the open state (open state in structure **12, 13, 14**) or when both substrates (CADA and αKG) are bound (structure **15**), R122 swings towards the active site and is in an inward conformation, ready for αKG binding. The positioning of R122 is similar to that of R157 of ScoE when tartrate is bound (Fig. 5C).

The structural overlay of the WT Rv0097 structures **1, 2, 14, 15, 16** revealed three distinct His loop conformations (H1, H2, and H3) and two distinct Arg loop conformations (R1 and R2), as represented by the

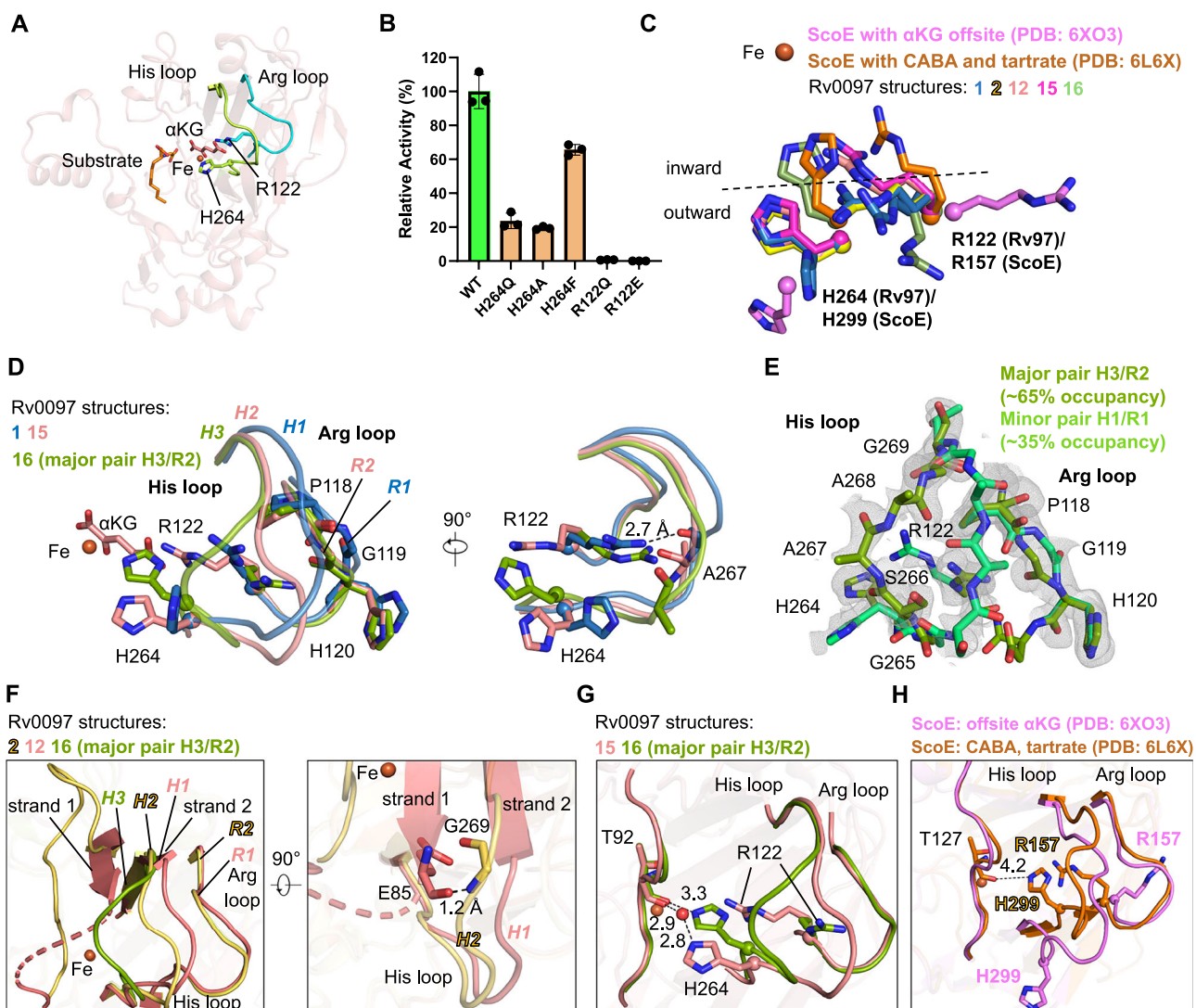

**Fig. 5 | Two surface loops create an access channel to the active site.**
**A** Locations of two loops on the protein surface. The His loop is shown in green and the Arg loop in cyan. H264 and R122 side chains are shown in sticks. CADA is shown in orange sticks. Fe ion is shown in orange sphere. **B** Activity assay of H264 and R122 variants showing relative product formations compared to the WT enzyme. For each assay, data and error bars represent average and standard deviation from three independently performed experiments, respectively. Source data are provided as a Source Data file. **C** H264 and R122 side chain rotamers of WT structures reported in this and previous works can be classified as inward and outward conformations. **D** Overlay of His and Arg loops from structures **1**, **15**, and **16** (with major conformation of His and Arg pair shown). The three distinct His loop conformations are labeled as H1, H2 and H3. The two distinct Arg loop conformations are labeled as R1 and R2. **E** Two discrete loop pair conformations H3/R2 (major conformation in olive green) and H1/R1 (minor conformation in light green) observed in structure **16** shown in a 1.0 σ 2mF$_o$-DF$_c$ composite omit map. **F** Conformational rearrangements of the "right lid" lock the His and Arg loops in place in H1 and R1 conformations. (left) Steric clashes between the major conformation H3 in **16** and strand 1 of the open conformation of **12**; (right) steric clashes between H2 in **2** and strand 1 in **12**. Residue G269 of **2** clashes with E85 in **12**. **G** The inward position of H264 in structure **15** hydrogen bonds to the neighboring loop backbone through an intermediate water molecule, whereas in the H3/R2 loop pair in **16**, H264 seals up the substrate pocket by direct contact with the neighboring loop backbone. **H** Conformational changes of the two loops and the H299/R157 side chains observed in two ScoE structures.

three structures shown in Fig. 5D. The H1 and H2 conformations of the His loop and both Arg loop conformations are observed in two chains of the apo structure (Supplementary Fig. 18). Closer inspections of H264/R122 side chain positions in these structures and their corresponding loop conformations reveal correlated motions. When the His loop is in H1 conformation (nearest to the Arg loop), the Arg loop is in the R1 conformation (Fig. 5D, left). R122 side chain is in outward conformation to form a 2.7 Å hydrogen bond with backbone carbonyl oxygen of A267 (Fig. 5D, right). When the His loop moves to the left (H1 to H2), the Arg loop moves to the left with it (R1 to R2). For the Arg loop, going from R1 to R2 involves backbone peptide flips of P118 and G119 (Fig. 5D). As the side chain of R122 moves outward, the His loop

moves accordingly to prevent the steric clashes between R122 side chain and backbone carbonyl of A267 (Fig. 5D).

Interestingly, we observe two discrete His loop conformations in the same chain of our vanadyl and CADA bound structure (structure **16**). In addition to showing a H1 conformation, we find a H3 conformation not previously observed in other structures (Fig. 5D). The Arg loop is also in two discrete conformations (R1 and R2) that differ primarily by backbone flips at P118 and G119. Refined occupancies of the loops suggest that there is a major loop pair H3/R2 (~ 0.65 occupancy, olive green in Fig. 5E) and a minor loop pair H1/R1 (~ 0.35 occupancy, light green in Fig. 5E). In the major H3/R2 pair, H264 is pointing inward and R122 outward, and in the minor pair, H264 has

flipped and is now pointing outward whereas R122 has changed positions but is still outward.

The His loop is the extension of strand 2 that is part of the "right lid," which undergoes conformational changes so that CADA can bind, and products can leave. As the "lids" open, the Arg loop moves from R2 to R1 conformation, and the His loop moves from H2 to H1 position and R122 becomes positioned for αKG binding (Fig. 5F, left). The series of loop movements is a result of the loop-to-strand secondary structure change of strand 1 in the open state. Strand 1 in the open state (pink in Fig. 5F, right) would have steric clashes with His loop when it is in H2 conformation (1.2 Å distance between E85 of strand 1 in open state and G269 of His loop in H2 conformation) (Fig. 5F, right); therefore, H2 to H1 conformational change is necessary. Arg loop thus moves accordingly from R2 to R1 conformation.

In the vanadyl and CADA bound structure **16**, the conformation of the major H3/R2 pair creates a space in between the two loops large enough for the long side chain of R122 to swing out to its outward conformation without clashes with His loop backbone (Supplementary Fig. 19). We hypothesize that this opening could also serve as an access channel for the dissociation of αKG-reaction products, succinate and $CO_2$, following the first half reaction, and the association of a second αKG for the second half-reaction. R122 side chain being in the most extended outward conformation makes it easier to bring αKG into the Fe site without having to fully open the enzyme (left and right "lids" in open conformation) during the catalytic cycle, thereby keeping CADA-intermediates protected. When the enzyme is in the open state, His/Arg loops are locked in position (the H1/R1 conformation), so they are unable to form the H3/R2 access channel, but this access channel is not necessary when the enzyme is in the open state. When open, the active site is readily accessible for co-substrate entry and co-product exit as is required at the start of turnover and at the end of the second half reaction when turnover is complete.

H264 seems to play a role in securing the various loop conformations so that αKG and its cleavage products can come and go in between the two half reactions without opening up the active site to solvent, protecting the CADA-intermediate. When αKG is bound to an inward pointing R122 (structure **15**), the H264 side chain hydrogen bonds to the backbone of the adjacent loop (backbone of T92) via an intermediary water molecule that is close to the active site (Fig. 5G), stabilizing the loop conformations and preventing solvent from entering the active site. When succinate and $CO_2$ leave and R122 points outward (major loop pair in structure **16**), as is expected in between two half-reactions, the H264 side chain directly forms a 3.3 Å hydrogen bond with T92 backbone carbonyl, further sealing up the reaction site and protecting reactive intermediate species Fig. 5G). H264 can only be in this most inward position when R122 points outward, since the two residues would sterically clash if both were pointed inward (green H264 and red R122 in Fig. 5G). Overall, these observed conformational changes of H264/R122 side chains and the His and Arg loops are similar to what was reported for ScoE (Figs. 5G and 5H), but now we know that these changes are related to αKG binding and are not an artifact of tartrate in the ScoE structure.

### Imine-CADA intermediate appears to stay bound to enzyme between half reactions

To investigate whether the imine-CADA intermediate remains sequestered within the enzyme active site between two half-reactions, we monitored imine-CADA formation in Rv0097-catalyzed reactions employing our previously established protocol[14]. Here we derivatized an acid-catalyzed degradation product of imine-CADA, glyoxylate, using 2-nitrophenylhydrazine (2-NPH)[14]. Our assays monitor imine-CADA production under three conditions: (1) full reaction condition, where the complete enzymatic reaction mixture was directly analyzed; (2) filtrate condition, where the enzyme was removed by centrifugation and only the filtrate was analyzed; and a no-enzyme control. After

1-minute incubation periods, we observed significantly higher glyoxylate production in the full reaction condition (Supplementary Fig. 20), suggesting that the imine-CADA intermediate stay bound to the enzyme during catalysis.

## Discussion

Enzymes in the non-heme Fe(II)/αKG-dependent dioxygenase family catalyze a vast assortment of challenging chemical transformations, such as hydroxylation, halogenation, desaturation, and demethylation reactions[36,41]. The recent discovery of ScoE from *Streptomyces coeruleorubidus* in INLP biosynthesis expanded their catalytic repertoire and showcased their capability of isonitrile formation[11]. We characterized the ScoE-catalyzed reaction and proposed a two half-reaction mechanism for isonitrile synthesis involving two Fe(IV)-oxo intermediate-catalyzed oxidation reactions[14]. This mechanism presents several challenges to ScoE's enzyme design, and our structural work showed that the αKG-binding site was not always available and appeared to be transiently induced. However, before this work, no structures of ScoE or its homologs were available with αKG bound in a catalytically competent conformation, limiting our understanding of the regulation mechanism of the enzyme.

In this work, we set out to perform detailed structural studies on Rv0097, a ScoE homolog from *Mtb*, to uncover the mechanism for substrate binding and specificity. Our previous work identified G204 as a gating residue that determines Rv0097's substrate specificity[12]. Consistently, recent bioinformatic and structural studies[26,42] of Rv0097 identified residues in the substrate-binding pocket (A202, T203, G204) that could contribute to substrate specificity by differentiating substrates with respect to their alkyl chain lengths (PDB: 8KHT). Incorporating bulky residues at these identified positions shifted the substrate specificity toward smaller alkyl chains. Here, we unveiled an additional layer in substrate specificity regulation–the role of F102/G204 residue pair in governing substrate binding and specificity. We showed that F102 exists in two rotamer conformations that keep the substrate-binding pocket hydrophobic, and that substitutions of F102 perturb the hydrophobicity of this pocket. Our structures additionally revealed a role of T203 in regulating solvent access to the CADA tail binding pocket. We envision that Rv0097 and its homologs could be engineered to perform isonitrile biosynthesis on other glycyl-fatty acid substrates with hydrophilic fatty acyl motifs, such as hydroxy fatty acids, by F102 substitutions combined with alterations of other residues in the substrate-binding pocket. These structural data should facilitate engineering efforts to expand the substrate scope of Rv0097 and homologs.

Additionally, this work has yielded insight into how Rv0097's two half-reactions are regulated. We have complemented the recently published Rv0097 structure[26] with 16 distinct structures and accompanying biochemical data. Collectively, these data have allowed us to propose how the enzyme's exquisite control of its two half-reactions is enabled by choreographed conformational changes across multiple sites (Fig. 6). First, our apo structure represents a resting state without relevant substrates bound. F102 is in two rotamer conformations and H264/R122 point away from the active site (Resting state in Fig. 6). Next, Rv0097 opens for αKG and CADA binding. In the open conformation, both H264/R122 point toward the active site and F102 is in its second rotamer conformation (F102′) (Open state in Fig. 6). Whether the Rv0097 reaction is ordered or random in binding αKG and CADA has not been established. In Fig. 6, we show αKG binding before CADA as soaking crystals in 5 mM each of αKG and CADA, resulted in an αKG-bound structure (**14**). Structure **14** shows one end of αKG binding bidendate to Fe, as is found in all other family members, and the other end of αKG interacting with an inward orientation of R122 (Open state, αKG-bound in Fig. 6). Following CADA binding, the active site closes for radical-based chemistry (Ternary complex in Fig. 6). $O_2$ will react with αKG, forming the putative Fe(IV)oxo-species and

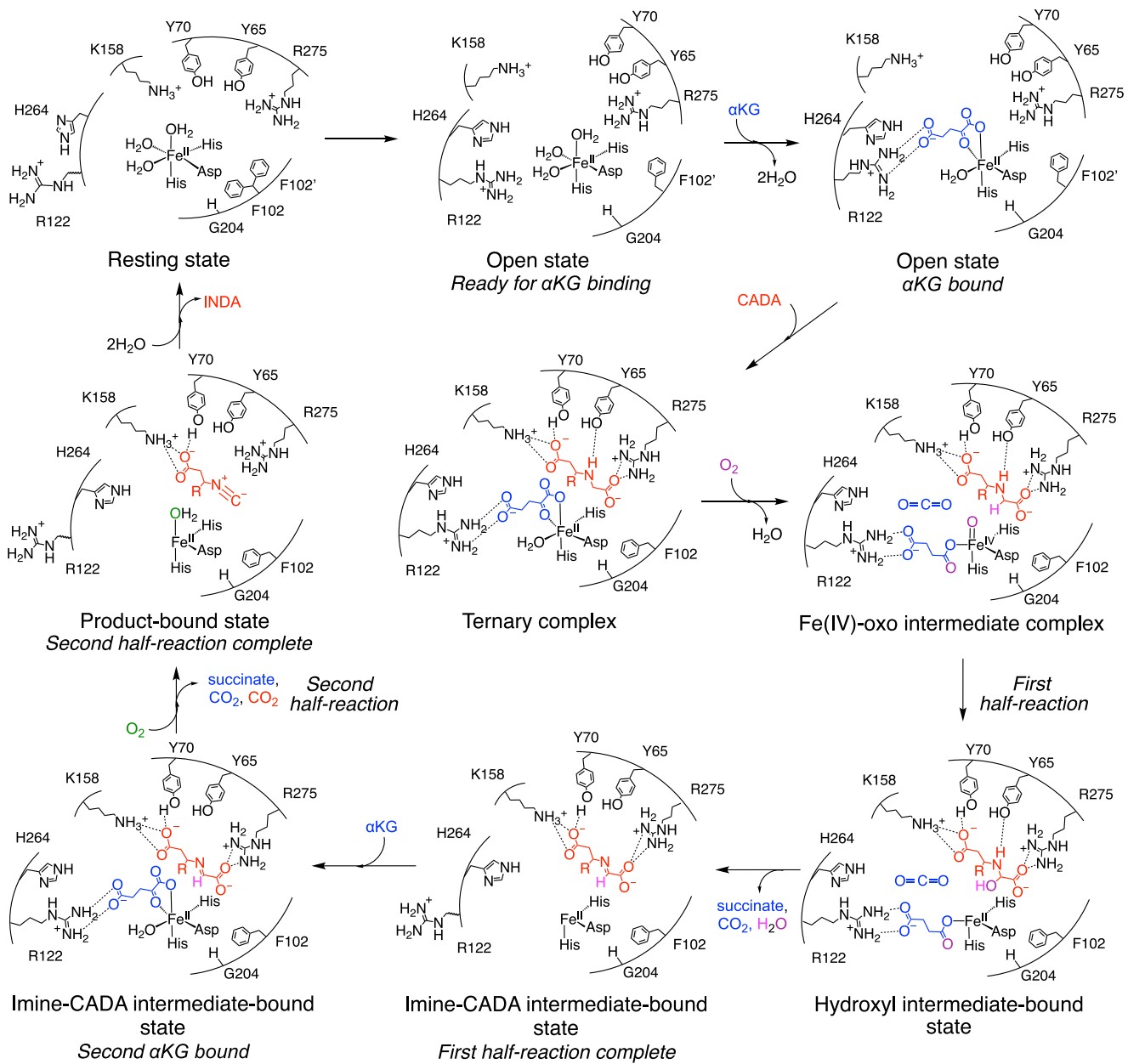

**Fig. 6 | Proposed structural mechanism for Rv0097-catalyzed isonitrile formation.** R = $C_7H_{15}$. Charges of amino acid side chains are assigned based on their p$K$a at physiological pH ( - 7.4).

succinate and $CO_2$ (Fe(IV)-oxo intermediate complex in Fig. 6). The Fe(IV)-oxo species would then perform a hydrogen atom abstraction on CADA, followed by hydroxylation to form the hydroxyl-CADA intermediate (Hydroxyl intermediate-bound state in Fig. 6). R122 now swings out, and co-products succinate and $CO_2$ will be released through the channel created in between the His and Arg flexible loops (H3/R2 configuration) together with loss of the hydroxyl group as a water molecule (see Fig. 1C), forming the imine-CADA intermediate (Imine-CADA intermediate-bound state in Fig. 6). At this point, the first half-reaction is complete. The second αKG enters through the same channel, interacts with R122, which adopts an inward orientation, and binds bidentate to the Fe center (Imine-CADA intermediate-bound state, second αKG bound in Fig. 6). The second half-reaction is then initiated upon binding of the second $O_2$ to Fe(II), which reacts with αKG forming succinate, $CO_2$, and a reactive Fe(IV)-oxo species. The Fe(IV)-oxo species abstracts a hydrogen atom from the imine-CADA intermediate, causes decarboxylation (see Fig. 1C). Thus, two $CO_2$

molecules are formed in the second half-reaction: one $CO_2$ (in blue) is derived from αKG, and the other $CO_2$ (in red) from CADA. Product INDA is formed. (Product-bound state in Fig. 6). Finally, the product INDA is released, and the enzyme returns to its resting state (Resting state in Fig. 6).

ScoE and Rv0097 are not alone in catalyzing multiple, consecutive oxidation reactions in the mechanism. Numerous multifunctional non-heme Fe(II)/αKG-dependent dioxygenases that catalyze sequential oxidation reactions are reported in fungal secondary metabolites biosynthesis[9,22–24]. However, these fungal enzymes and ScoE/Rv0097 employ different mechanisms to regulate the reactions they catalyze. In a recent structural study on two highly homologous bifunctional fungal Fe(II)/αKG enzymes, AusE and PrhA, the authors proposed that their oxidized product from the first half-reaction might leave the active site with succinate and re-enter upon the binding of the second αKG[43]. Notably, this departure and re-binding mechanism is plausible for PrhA since Berkeleyone B, the intermediate formed after one round

of oxidation, is stable and can be isolated naturally from the extremophilic fungus *Penicillium rubrum*[44]. However, this intermediate-rebinding mechanism is less likely for Rv0097 since its imine-CADA intermediate is not as stable. Also, the derivatization assay indicates that imine-CADA remains bound to Rv0097. The flexible His and Arg loops observed in the Rv0097 structures seem ideally suited to allow succinate departure and αKG re-binding to be uncoupled from the departure of the CADA-derived intermediate. These structural differences suggest distinct regulatory mechanisms for fungal multifunctional non-heme Fe(II)/αKG dioxygenases and Rv0097, a bacterial bifunctional non-heme Fe(II)/αKG dioxygenase, in catalyzing iterative and sequential oxidation reactions.

In this work, we provide a detailed structural framework of a highly dynamic non-heme Fe(II)/αKG-dependent isonitrile synthase, Rv0097. To synthesize an isonitrile group starting from a glycyl moiety is challenging, but nature figured out a way to perform this demanding reaction – by breaking it down into two steps and generating and using a highly reactive iron-oxygen intermediate twice. It is impressive that nature developed a macromolecular machine with features tailored to allow for the exquisite control of the half-reactions. From a flipping single amino acid, to altering flexible loop positions, to moving "lid" regions of the enzyme, enzyme structural dynamics has been leveraged to create a metalloenzyme that can single-handedly perform two radical-based half reactions without having to release into solution a reactive or unstable reaction intermediate. Whereas the structural and biochemical data presented here provide a basis for how this bifunctional enzyme can catalyze multiple oxidation reactions in one catalytic cycle, there is still more to learn about the reaction mechanism itself. For example, in all our αKG-bound structures, αKG binds in its offline conformation, occupying the site where an online Fe(IV)-oxo intermediate is expected. Such a conformation raises the question of whether molecular oxygen binding to Fe(II) promotes αKG rearrangement from offline to online configuration or if rearrangements occur after the Fe(IV)-oxo intermediate is generated. We suspect that these remarkable enzymes will continue to fascinate for some time to come. Nature has already created enzyme variants in the form of ScoE, MmaE, and Rv0097 that have different substrate specificity[12] and generate different natural product building blocks, indicating that the potential for engineering these enzymes to create an even greater chemical repertoire of building blocks is substantial.

## Methods

### Commercial materials

Q5 High-Fidelity PCR Master Mix (New England Biolabs) was used for polymerase chain reactions (PCR). Restriction enzymes were purchased from New England Biolabs. Oligonucleotides were purchased from ELIM BIOPHARM. All chemicals used in this work were purchased from Sigma-Aldrich, Fisher Scientific, Alfa Aesar, or Hampton Research, unless otherwise noted. 3-aminodecanoic acid hydrochloride (purity: 95%) was purchased from Enamine.

### Construction of plasmids for expression in *E. coli*

The *rv0097* gene was PCR amplified from genomic DNA (*Mycobacterium tuberculosis* H37Rv) and cloned into pET24b+ through restriction enzyme digestion (ThermoFisher) and ligation with NEBuilder HiFi DNA Assembly (New England Biolabs). The primers and vectors used in this study are reported in Supplementary Table 3 and 4, respectively. Plasmids were extracted using a Zyppy Miniprep Kit (Zymo Research) and confirmed by DNA sequencing at ELIM BIOPHARM.

### Site directed mutagenesis

Rv0097 variants were constructed using the Agilent QuickChange II Site Directed Mutagenesis kit by following the manufacturer's protocol with the primers listed in Supplementary Table 3. pET24b + -Rv0097

was utilized as a template for the PCR reactions. The introduction of point mutations was confirmed with DNA sequencing and the corresponding Benchling link to the sequencing data are listed in Supplementary Table 4. *E. coli*. strains used are listed in Supplementary Table 5.

### Protein expression and purification for assays

The expression and purification of Rv0097 was conducted using a similar procedure as previously reported[12,45]. BL21 Star (DE3) competent cells were inoculated (2% inoculum ratio) to 1 L of Terrific Broth (TB) in a shake flask containing 50 μg/mL of kanamycin. The cultures were grown at 37 °C at 250 RPM to an $OD_{600}$ of 0.6. The cells were then cooled on ice for 10 min and induced with 250 μM isopropyl-β-D-galactopyranoside (IPTG) for 16 h at 16 °C and 250 RPM. Cells were harvested by centrifugation (6,371 x g, 15 mins, 4 °C), resuspended in 30 mL of lysis buffer (50 mM HEPES pH 8.0, 500 mM NaCl, 5 mM imidazole), and lysed by sonication on ice. Cellular debris was removed by centrifugation (27,216 x g, 45 min, 4 °C), and the supernatant was filtered with a 0.45 μm filter before batch binding. Nickel-Nitrilotriacetic acid (Ni-NTA) resin (Qiagen) was added to the filtrate at 3 mL/L of culture, followed by incubation for 1 h at 4 °C. The protein-resin mixture was loaded onto a gravity flow column in which the flowthrough was discarded. The column was then washed with approximately 30 mL of wash buffer (50 mM HEPES pH 8.0, 100 mM NaCl, 20 mM imidazole). Proteins were eluted in approximately 15 mL of elution buffer (50 mM HEPES pH 8.0, 100 mM NaCl, 250 mM imidazole). The entire process was monitored with a Bradford assay. Rv0097 was concentrated to 10 mL using a 10 kDa Amicon spin filter and subsequently dialyzed using a 10 kDa Slide-A-Lyzer cassette overnight in 2 L of dialysis buffer (50 mM HEPES pH 8.0, 100 mM NaCl, 1 mM EDTA). The dialysis buffer was changed twice during the overnight period (every 2 h). Rv0097 was concentrated using a 10 kDa Amicon spin filter to 40 mg/mL (with 10% v/v glycerol). Rv0097 was flash frozen in liquid nitrogen and stored in 30 μL beads at −80 °C. The presence and purity of purified proteins were assessed using sodium dodecyl sulfate–polyacrylamide gel electrophoresis (SDS-PAGE). Protein concentration was determined by UV/Vis absorbance at 280 nm using an extinction coefficient of 45,380 $M^{-1}$ $cm^{-1}$ that was determined using the ProtParam tool[46].

The approximate molecular weight and protein yield of Rv0097 are as follows: 34 kDa, 10 mg/L from TB, Uniprot: P9WG83. The Rv0097 variants had purification yields that were nearly the same as that of the WT enzyme ( ± 8%).

### Reconstitution of apo-Rv0097 to yield holo-Rv0097 for assays

A Bio-Rad Bio-Gel P-6 gel column was equilibrated in exchange buffer (50 mM HEPES pH 8.0, 100 mM NaCl). Rv0097 was desalted to remove EDTA by following the manufacturer's protocol. The enzyme was immediately added to the biochemical assay and supplemented with ammonium iron (II) sulfate hexahydrate matching 90% of the protein concentration.

### Synthesis of CADA and CABA

In general, the syntheses of racemic CADA and enantiomerically pure CABA were followed as previously reported[12,45]. CADA was isolated as a colorless oily liquid and confirmed by high-resolution mass spectrum (HRMS) and nuclear magnetic resonance (NMR). NMR spectra were recorded with a Bruker AVANCE spectrometer at 900 MHz ($^{1}$H NMR), and 226 MHz ($^{13}$C NMR). NMR solvents were purchased from Cambridge Isotope Laboratories Inc. Other chemicals and solvents were purchased from Sigma-Aldrich. High-performance liquid chromatography (HPLC) purification was carried out using an Agilent 1200 system and a Grace Alltima C18 column (150 ×10 mm). LC-HRMS was conducted using an Agilent 6545 Accurate-Mass Q-TOF. CADA: calculated for $C_{12}H_{23}NO_4$ $[M + H]^+$: 246.1700; found 246.1695 (2 ppm

error).[1]H NMR (900 MHz, DMSO-$d_6$) δ 3.86 (s, 2H), 3.47 (m, 1H), 2.75 (dd, $J$ = 17.1, 5.1 Hz, 1H), 2.59 (dd, J = 17.1, 6.8 Hz, 1H), 1.71 (m, 1H), 1.55 (m, 1H), 1.30–1.20 (m, 10H), 0.86 (t, $J$ = 7.1 Hz, 3H). [13]C NMR (226 MHz, DMSO-$d_6$) δ 171.96, 168.34, 54.12, 45.06, 34.83, 31.12, 30.48, 28.67, 28.39, 24.37, 22.06, 13.94. CABA: calculated for $C_6H_{12}NO_4$ $[M + H]^+$: 162.0761; found 162.0762 (0.6 ppm error).[1]H NMR (900 MHz, DMSO-$d_6$): δ 3.91 (d 144.0, 2H), 3.54 (m, 1H), 2.84 (dd 16.6, 4.3, 1H), 2.50 (m, 1H), 1.25 (d 6.5, 3H). [13]C NMR (225 MHz, DMSO-$d_6$): δ 171.6, 168.4, 50.3, 44.8, 37.1, 16.2. Mestrenova version 9.0.1 was used to analyze NMR data. LC-HRMS data were analyzed using Agilent MassHunter Qualitative Analysis software 10.0.

### Activity assays and quantification of isonitrile

A 100 μL biochemical assay consisting of 50 mM HEPES pH 8.0, 1 mM CADA, 2 mM αKG, 100 μM apo-Rv0097, and 90 μM $Fe(NH_4)_2(SO_4)_2$ was incubated at room temperature for 10 min. The enzymatic reaction was quenched with 200 μL of 667 μM 3,6-di(pyridine-2-yl)-1,2,4,5-tetrazine (Py-tetrazine) dissolved in cold methanol, gently mixed, and incubated for 1 h at room temperature. The quenched reaction mixture was further vortexed and centrifuged for 10 min to remove aggregated protein. LC-HRMS analysis was conducted on an Agilent Technologies 6545 Accurate-Mass QTOF LC-MS instrument and an Eclipse Plus C18 column (100 ×4.6 mm). Chromatography was performed using a linear gradient of 10-50% acetonitrile (v/v) with 0.1% formic acid over 12 min in water with 0.1% formic acid at a flow rate 0.5 mL/min. The production of INDA was estimated by looking for the formation of 3,5-di(pyridine-2-yl)-1Hpyrazol-4-amine (Py-AP) and comparing its retention time and mass spectrum with a standard. The Py-AP standard was synthesized and characterized by NMR[47] was observed with $[M + H]^+$: 238.1088 (calculated $[M + H]^+$: 238.1086, 0.8 ppm error). Negative controls omitting CADA, αKG, and Rv0097 were also conducted. LC-HRMS data were analyzed using Agilent MassHunter Qualitative Analysis software 10.0.

### Determination of kinetic parameters of Rv0097 towards CADA

100 μL biochemical assays were performed in triplicate containing 50 mM HEPES pH 8.0, 5 mM αKG, 10 μM apo-Rv0097, and 9 μM $Fe(NH_4)_2(SO_4)_2$. The reactions were initiated by adding αKG and CADA was supplied at varying concentrations depending on the Rv0097 variant in question. After incubation, the enzymatic reaction was quenched with 200 μL of 667 μM Py-tetrazine dissolved in cold methanol, gently mixed, and incubated for 1 h at room temperature. The subsequent reactions were centrifuged to remove protein debris, and the supernatant was analyzed with LC-HRMS. Time points were taken at 30 s, 1 min, 2 min, 5 min, and 10 min to determine the initial velocity of INDA formation. The product concentration was estimated by constructing a standard curve of Py-AP by analysis of standards containing 100 μL of 50 mM HEPES pH 8.0 and varying amounts of Py-AP that were subsequently quenched with 200 μL of 667 μM Py-tetrazine dissolved in cold methanol. Chromatography was performed using a linear gradient of 10-50% acetonitrile (v/v) with 0.1% formic acid over 12 min in water with 0.1% formic acid at a flow rate 0.5 mL/min. Kinetic parameters were then calculated using GraphPad Prism 9. LC-HRMS data were analyzed using Agilent MassHunter Qualitative Analysis software 10.0.

### Estimation of Rv0097-bound imine-CADA versus free

A 200 μL assay containing 50 mM HEPES pH 8.0, 1 mM CADA, 1 mM αKG, 50 μM apo-Rv0097, and 45 μM $Fe(NH_4)_2(SO_4)_2$. The reaction was split into two and performed as described[14]. Briefly, reactions were quenched after short incubation periods (1 min), followed by chemical derivatization to assess glyoxylate formation for imine intermediate quantification. To distinguish between free and enzyme-bound imine, the relative production from the reaction supernatant alone (enzyme is removed via filtration) to that of the complete reaction mixture containing the enzyme. A no enzyme control was also performed. Results are shown in Supplementary Fig. 20).

### Protein expression and purification for crystallization

To aid soluble expression and maximize protein yield, Rv0097 WT and its G204F, F102Y, F102W, and F102A variants were co-expressed with molecular chaperones using an auto-induction protocol, as follows. 1.0 μL of 10 ng/μL pET24b + -Rv0097 plasmid (see above for plasmid construction) was mixed with 1.0 μL of 10 ng/μL chaperone plasmid pGro7[48] (Takara Bio) that contains *groES*-*groEL* under *araB* promoter and chloramphenicol antibiotic resistant marker. The resulting 2.0 μL plasmid mixture was co-transformed into BL21 Star (DE3) Chemically Competent *E. coli* (Invitrogen, ThermoFisher). A single colony was picked to inoculate an overnight starter culture (16 hrs) in 50 mL LB medium containing 50 μg/mL kanamycin (GoldBio) and 25 μg/mL chloramphenicol (OmniPur, EM Science), with 220 RPM shaking at 37 ⁰C. 5 mL of the overnight starter culture was used to inoculate 1 L of autoclaved auto-induction media (prepared following Studier, F. W[49].) containing 150 μg/mL of kanamycin and 25 μg/mL chloramphenicol. 2.0 g of L-(+)-arabinose (Sigma-Aldrich) was added to the 1 L culture during inoculation to allow for chaperone expressions prior to Rv0097 expression. Cells were grown with 220 RPM shaking at 37 ⁰C until $OD_{600}$ reached ~0.6. Cell cultures were taken out of the incubator and cooled at 4 ⁰C for 30 min without shaking, after which they were put back to the incubator for an additional 20 h with 220 RPM shaking at 22 ⁰C for protein expression. Cells were harvested by centrifugation (9000 x $g$, 10 mins, 4 ⁰C), and flash-frozen into liquid nitrogen before storage at −80 ⁰C until cell lysis and protein purification.

Cells from 1 L culture were taken out of −80 ⁰C and let thaw under 4 ⁰C, then resuspended in 50 mL of lysis buffer (50 mM HEPES pH 8.0, and 500 mM NaCl) supplemented with 1 mg/mL egg-white lysozyme (GoldBio), 0.15 μL/mL benzonase nuclease (Sigma-Aldrich), and 1 tablet of cOmplete EDTA-free protease inhibitor cocktail (Sigma-Aldrich), and put on a nutator for 1 h at 4 ⁰C to allow complete thawing and resuspension. Cell lysis was completed by ultrasonication on ice using the following parameters: 60% amplitude, 8 cycles of 30 s, 1 s on 2 s off per cycle. The resulting suspension was clarified by centrifugation (28,000 x $g$, 50 min, 4 ⁰C). The clarified lysate was filtered using 0.45 μm filter before loading onto a 5-mL HiTrap TALON crude column (Cytiva) pre-equilibrated with 10 column volumes (CV) of lysis buffer at 5 mL/min flow rate, followed by 30 min of batch-binding to ensure maximized protein binding. The column was then washed with 25 CV of wash buffer (50 mM HEPES pH 8.0, 300 mM NaCl, and 5 mM imidazole) and eluted with 10 CV of elution buffer (50 mM HEPES pH 8.0, 300 mM NaCl, and 150 mM imidazole). The eluted protein was concentrated using a 10 kDa MWCO (Amicon) centrifugal filter to a final volume of 2.0 mL, then buffer-exchanged by dialysis under 4 ⁰C against 800 mL of dialysis buffer (50 mM HEPES pH 8.0, 300 mM NaCl, 1 mM acid EDTA) using a 10 KDa MWCO Slide-A-Lyzer dialysis cassette (ThermoFisher). The dialysis buffer was changed once after 4 h before overnight dialysis (~18 h). Buffer-exchanged protein was further purified by a 1 CV run of size-exclusion chromatography (SEC) by loading onto a Superdex 200 16/60 size-exclusion column pre-equilibrated with 2 CV of SEC buffer (25 mM HEPES pH 8.0, and 100 mM NaCl). Fractions correspond to Rv0097 dimer were pooled and concentrated using a 10 kDa MWCO (Amicon) centrifugal filter until a final concentration of 10.0 mg/mL (0.3 mM). Throughout purification, protein purity was monitored by 4 to 20 % (w/v) SDS-PAGE. C-terminal purification tag (amino acid sequence: KLAAALEHHHHHH) was not cleaved. Protein concentration was determined by UV/Vis absorbance at 280 nm using an extinction coefficient of 45,380 $M^{-1}$ $cm^{-1}$ that was determined using the ProtParam tool[46]. Concentrated protein was either immediately used for crystallization (see below), or aliquoted and flash-frozen in liquid nitrogen then stored at −80 ⁰C for future use.

## Crystallization

Rv0097 WT purified as described above was used directly to identify initial crystallization conditions using PEG/Ion HT crystallization screen (Hampton Research) dispensed by an NT8 crystallization robot (Formulatrix) to 96-well sitting-drop trays at room temperature. The identified conditions were further optimized using hanging-drop vapor diffusion method at room temperature. All chemicals used for crystallization were purchased from Hampton Research. Crystals of apo-Rv0097 **1** were obtained by mixing 1.0 μL of apo Rv0097 and 1.0 μL of well solution (270 mM magnesium acetate and 16% w/v PEG 3,350), in a sealed well over 500 μL of well solution. Protein crystals typically appeared in 3–5 days and continued to grow over 2-3 weeks. After 2 weeks, crystals were harvested, cryoprotected with paraffin oil, and flash frozen in liquid nitrogen, leading to the apo dataset for Rv0097.

Several drops containing crystals of apo-Rv0097 were used to create a seed stock using seed bead kit (Hampton Research). Briefly, drops were combined and vortexed for 3 min total, with one-minute breaks on ice for every 30 s of vortexing. For long-term storage, prepared seed stocks were flash frozen in liquid nitrogen and stored at −80 $^{0}$C until next use. The seed stock and its serial dilutions (up to 100,000-fold dilution) were used for subsequent micro-seeding crystallizations. To obtain the apo-G204F **3**/F102Y **6**/F102W **8**/F102A **10** structures, aerobically purified protein (10.0 mg/mL, 0.3 mM) was directedly used for crystallization using diluted seed stocks. The hanging drops consisted of 1.0 μL of Rv0097 variant and 1.0 μL of diluted seed stock, in a sealed well over 500 μL of well solution. Crystals grew out of micro-seeds typically appeared in 2 days (crystals appeared overnight in drops with more concentrated seed stock) and continued to grow to full size in 1–2 weeks. After 2 weeks, these crystals, and all crystals described below, were harvested, cryoprotected with paraffin oil, and flash frozen in liquid nitrogen.

To obtain Fe-CADA Rv0097 co-crystals **2**, frozen aliquots of purified Rv0097 (10.0 mg/mL, 0.3 mM) were brought into an MBraun anaerobic chamber in an $N_2$ environment and let thaw on cold beads. After thawing, the protein aliquots were allowed to equilibrate on cold beads for at least 30 mins with PCR tubes open before adding 1 molar equivalence (0.3 mM) of Fe(NH$_4$)$_2$(SO$_4$)$_2$, 5 molar equivalence (1.5 mM) of CADA (see above for its synthesis), and 5 molar equivalence (1.5 mM) of αKG (α-Ketoglutaric acid disodium salt dihydrate, Sigma-Aldrich) (1.5 mM), then gently mixed and incubated overnight at 4 $^{0}$C. The hanging drops consisted of 1.0 μL of Fe- and CADA- supplemented Rv0097 and 1.0 μL of diluted seed stock, in a sealed well over 500 μL of well solution. Crystals started to appear in 1–2 days and grew to full size within 2 weeks. These crystals were then transferred to a Coy anaerobic chamber with an Ar/N$_2$ gas mix environment for looping. Crystals were then harvested, cryoprotected with paraffin oil, and flash frozen in liquid nitrogen. To obtain the ternary complex structure **15**, the Fe-CADA Rv0097 co-crystals were looped and deposited into a 3.0 μL drop of soaking solution that contains 10 mM αKG and 16% w/v PEG 3,350 for 1 h prior to looping. For anaerobic looping, the soaked crystals were transferred to a Coy anaerobic chamber with an Ar/N$_2$ gas mix environment for cryoprotection and cryo-cooling.

To obtain the CADA- and/or αKG- soaked crystals, frozen aliquots of purified Rv0097 (10.0 mg/mL, 0.3 mM) were brought into an MBraun anaerobic chamber in an N$_2$ environment and let thaw on cold beads. After thawing, the protein aliquots were allowed to equilibrate on cold beads for at least 30 mins with PCR tubes open before adding 1 molar equivalence (0.3 mM) of Fe(NH$_4$)$_2$(SO$_4$)$_2$ then gently mixed and incubated overnight at 4 $^{0}$C. The hanging drops consisted of 1.0 μL of Fe- and CADA- supplemented Rv0097 and 1.0 μL of diluted seed stock, in a sealed well over 500 μL of well solution. Crystals started to appear in 1-2 days and grew to full size within 2 weeks. Then, the crystals were looped and deposited in 3.0 μL drops of soaking solutions and soaked for 1–2 h over sealed wells prior to looping. The soaking solutions contained the following: (1) 5 mM CADA and 16% w/v PEG 3350 for the

acetate-bound open conformation structure (5 mM CADA soaked structure **12**); or (2) 10 mM CADA and 16% w/v PEG 3350 for the CADA-bound open conformation structure (10 mM CADA soaked structure **13**); or (3) 5 mM CADA, 5 mM αKG, and 16% w/v PEG 3350 for the CADA- and αKG-bound open conformation structure (5 mM CADA and 5 mM αKG soaked structure **14**). Crystals were looped and cryo-cooled in a Coy anaerobic chamber using the conditions described above.

The Vanadyl-bound Rv0097 crystals were grown in aerobic conditions. To obtain the vanadyl-CADA WT co-crystals **16**, vanadyl-CADA G204F **4**/F102Y **9**/F102W **7**/F102A **11** variants co-crystals, and vanadyl-CABA-G204F **5** co-crystals, aerobically purified protein was directly supplemented with 1.3 molar equivalence (0.4 mM) of Vanadium(IV) oxide sulfate hydrate (Sigma-Aldrich), 5 molar equivalence (1.5 mM) of CADA (or CABA for the vanadyl-CABA-G204F structure **5**), and 5 molar equivalence (1.5 mM) of succinate (succinic acid, pH 7.0, Hampton Research), and incubated on ice for 2 h. The resulting protein solution was used for crystallization using diluted seed stocks. Crystals started to appear in 1–2 days and grew to full size within 2 weeks. Crystals were looped and cryo-cooled aerobically using conditions described above.

## X-ray data collection and processing

X-ray diffraction data were collected at cryogenic temperature (100 K) at the Stanford Synchrotron Radiation Lightsource (SSRL) (Menlo Park, CA) beamline 9–2 using a Dectris Pilatus 6 M detector. All data were indexed, integrated, and scaled in XDS, with data collected at the Vanadium peak wavelength processed anomalously with Friedel pairs kept separate. Resolution cutoffs were determined based on collective considerations of R$_{sym}$, CC$_{1/2}$, I/σ, and data completeness in the highest resolution bin. Data collection statistics are summarized in Supplementary Table 1.

## Structure determination and refinement

The Rv0097 WT apo structure **1** was solved by molecular replacement (MR) using the crystal structure of Rv0097 homolog from *Streptomyces coeruleorubidus*, ScoE[13] (45% identity, 61% similarity), as the search model (PDB ID: 6XOJ). Prior to MR, the model was improved in phenix.sculptor[50] by gap-based mainchain deletion (sequence alignment-based deletion of residues present in the search model but not in the target) and Schwarzenbacher-style[51] sidechain pruning. Edited search model was then used for MR in phenix.Phaser-MR[52] using data extended to full resolution. The final MR solution with LLG of 810.6 and TFZ 29.9 identified two protomers in the asymmetric unit (ASU), corresponding to a Rv0097 dimer, consistent with our SEC data that Rv0097 exists as a homodimer in solution. This final MR solution was refined in phenix.refine[53] first using rigid-body refinement, with 5% of total reflections randomly selected and set aside for the test set (R$_{free}$). One round of rigid-body refinement led to an initial R$_{free}$ of 0.49. Several following rounds of positional refinement and group B-factor refinement were sufficient to reduce R$_{free}$ to 0.44. Iterative rounds of positional and individual B-factor refinements were then performed, with manual adjustments in Coot[54] against 2mF$_o$-DF$_c$ maps contoured to 1.0 σ. Two-fold non-crystallographic symmetry (NCS) restraints were used throughout the refinements. A 2mF$_o$-DF$_c$ simulated-annealing composite omit map was generated with 5% of the model omitted for individual omit maps, using Phenix Composite_omit_map[55] after each round of refinement to verify this structure and all structures described below. Water and other solvent molecules from crystallization conditions were manually added to this structure, and to all structures described below, using mF$_o$-DF$_c$ map contoured to 3.0 σ and the composite omit map contoured to 1.0 σ as criteria. All 2mF$_o$-DF$_c$ and mF$_o$-DF$_c$ maps are generated by phenix.refine.

To obtain other structures in the same $P2_1$ space group with two molecules per ASU (structures **2-7, 9, 15, 16**) (see Supplementary Table 1-2), this test set from the apo dataset was carried over and used

in refinement. After one round of rigid-body refinement using the apo structure (structure **1**), iterative rounds of positional and individual B-factor refinements were then performed, with manual building and adjustments in Coot[54] against $2mF_o\text{-}DF_c$ and composite omit maps contoured to 1.0 σ. Structure and geometric restraints for the CADA and CABA molecules and the vanadyl ion were generated using eLBOW[56] using chemical SMILES strings (for CADA or CABA molecules) or the chemical component code VVO (for the vanadyl ion) as chemical inputs.

Since the F102Y apo structure **8** is in a different space group, *C2*, MR was used for phasing for this structure. Chain A of the refined apo-Rv0097 structure **1** was used for MR using Phaser-MR[52] without modifications to the model. The final MR solution with LLG of 9877.1 and TFZ of 85.9 identified one protomer per ASU. This MR solution was directly used for refinements, first using rigid-body refinement, with 5% of total reflections randomly selected and set aside for the test set ($R_{free}$). Following rigid-body refinement, iterative rounds of positional and individual B-factor refinements were then performed, with manual adjustments in Coot[53] against $2mF_o\text{-}DF_c$ maps contoured to 1.0 σ. Waters were added and the structure was validated as described above.

The F102A apo structure **10** and the F102A-vanadyl-CADA structure **11** are in another new space group, $P2_12_12$, therefore MR was used for phasing. Chain A of the refined apo-Rv0097 structure **1** was used for MR using Phaser-MR[52] without modifications to the model. For the F102A apo structure **10**, the final MR solution with LLG of 4580.2 and TFZ of 60.4 identified one protomer per ASU. This MR solution was directly used for refinements, first using rigid-body refinement, with 5% of total reflections randomly selected and set aside for the test set ($R_{free}$). Following rigid-body refinement, iterative rounds of positional and individual B-factor refinements were then performed, with manual adjustments in Coot[54] against $2mF_o\text{-}DF_c$ maps contoured to 1.0 σ. To obtain the F102A-vanadyl-CADA structure **11**, refined model of the F102A apo structure **10** was used directly for refinement, starting with rigid-body refinement. The test reflection set for F102A apo structure **10** was carried over and used in the refinements for the F102A-vanadyl-CADA structure **11**.

The rest of the CADA- and/or αKG- soaked structures **12, 13, 14** are in the $P2_1$ space group as most of the structures reported in this work, but their unit cell parameters are significantly different due to cell expansions caused by large conformational changes. Therefore, to obtain these structures, Chain A of the refined apo-Rv0097 structure **1** was used for MR using Phaser-MR[52] without modifications to the model. We started with the 10 mM CADA soak dataset **13**. The final MR solution with LLG of 16630.4 and TFZ 38.8 identified 4 protomers in the ASU. This MR solution was directly used for refinements, first using rigid-body refinement, with 5% of total reflections randomly selected and set aside for the test set ($R_{free}$). This test set was carried over and used in refinements of the 5 mM CADA soak structure **12** and the 5 mM CADA- and 5 mM αKG-soak structure **14**. Following rigid-body refinement, iterative rounds of positional and individual B-factor refinements were then performed, with manual adjustments in Coot[54] against $2mF_o\text{-}DF_c$ maps contoured to 1.0 σ. To obtain the 5 mM CADA soak structure **12** and the 5 mM CADA- and 5 mM αKG-soak structure **14**, refined model of the 10 mM CADA soak structure was used for refinement **13**, starting from rigid-body refinement, followed by iterative rounds of positional and individual B-factor refinements and manual adjustments.

A summary of the data collection and refinement statistics for all reported structures can be found in Supplementary Table 1. All structural figures were generated using the PyMOL Molecular Graphics software package (Schrödinger, LLC). Crystallography package was compiled by SBGrid Consortium[57].

## Computational channel calculations
Channel prediction was performed by MOLEonline[58] running MOLE version 2.5 using default parameters and with the atomic coordinates of structure **16**, minus the side chain of R122.

## Reporting summary
Further information on research design is available in the Nature Portfolio Reporting Summary linked to this article.

## Data availability
Atomic coordinates and structure factors for the crystal structures reported in this work have been deposited to the Protein Data Bank (PDB) under the following accession numbers: 9P9N (**1**), 9P9O (**2**), 9P9P (**3**), 9PAS (**4**), 9PAU (**5**), 9PAX (**6**), 9PAY (**7**), 9PBJ (**8**), 9PBN (**9**), 9PBU (**10**), 9PBX (**11**), 9PC5 (**12**), 9PCL (**13**), 9PCO (**14**), 9PCM (**15**), and 9PCN (**16**). Other PDB entries that were used for structural analyses in this work are: 8KHT, 6XO3, 6XOJ, and 6L6X. X-ray datasets for the corresponding crystal structures have been deposited to the SBGrid Data Bank under the following accession numbers: 1218 (**1**), 1220 (**2**), 1221 (**3**), 1222 (**4**), 1223 (**5**), 1224 (**6**), 1225 (**7**), 1226 (**8**), 1227 (**9**), 1228 (**10**), 1229 (**11**), 1230 (**12**), 1231 (**13**), 1232 (**14**), 1233 (**15**), 1234 (native dataset for structure **16**), and 1235 (V peak dataset for structure **16**). These datasets are available at https://data.sbgrid.org. UniProt accession number of Rv0097 sequence used in this study is P9WG83. A Source Data file is included to provide underlying data associated with biochemical experiments in this work. Other relevant data supporting the findings of this study are available in this article or its supplementary information. Source data are provided with this paper.

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

## Acknowledgements
This work was supported by NIH Grants R35 GM126982 to C.L.D. and R35 GM153289 to W.Z. C.L.D. is a Howard Hughes Medical Institute Investigator. A.D.R.F. was supported with a Stanford Science Fellowship. Use of the Stanford Synchrotron Radiation Lightsource (SSRL), SLAC National Accelerator Laboratory, is supported by the U.S. Department of Energy, Office of Science, Office of Basic Energy Sciences under Contract No. DE-AC02-76SF00515. The SSRL Structural Molecular Biology Program is supported by the DOE Office of Biological and Environmental Research, and by the National Institutes of Health, National Institute of General Medical Sciences (P30GM133894). This work utilized the Stanford Medicinal Chemistry Knowledge Center funded by the NIH High End Instrumentation grant (1 S10OD028697 – 01). The authors would like to thank Chaitan Khosla for useful discussions and advice on biochemical characterization of Rv0097. The contents of this publication are solely the responsibility of the authors and do not necessarily represent the official views of NIGMS or NIH.

## Author contributions
N.Y., A.D.R.F., W.Z and C.L.D. designed research; N.Y. expressed and purified enzymes and performed all crystallography experiments; A.D.R.F. conducted cloning, site-directed mutagenesis, expressed and purified enzymes, performed biochemical assays, kinetic characterization and chemical synthesis of CADA/CABA; N.Y., A.D.R.F., W.Z, and C.L.D. analyzed the data; N.Y. and C.L.D. wrote the paper; C.L.D. and W.Z. supervised the project.

## Competing interests
The authors declare no competing interests.
