## [Transparent Peer Review file · Nature Communications]

A highly dynamic mononuclear non-heme iron enzyme for the two-step isonitrile biosynthesis

Corresponding Author: Professor Catherine Drennan

Version 0:

Reviewer comments:

Reviewer #1

(Remarks to the Author)

The authors reported several high-resolution crystal structures of the isonitrile biosynthetic enzyme Rv0097 from *Mycobacterium tuberculosis*, including both wild-type and variant forms in the presence and absence of substrate (CADA) and α -ketoglutarate (α KG). These structural insights help elucidate the enzyme's architecture and suggest potential conformational changes that may contribute to isonitrile formation. However, the current manuscript lacks experimental evidence to substantiate the conclusions drawn from these structural snapshots. To this reviewer, it remains unclear whether the observed conformation differences correspond to catalytically relevant intermediates or they are simply artifacts during crystallization. In addition, several statements in the manuscript overlap with the authors' previous work and other published literatures. To merit publication in a high-impact journal such as *Nature Communications*, the authors should consider carrying out experiments to provide insights into the proposed mechanism. Specific comments are listed below:

1. For structures 3 (PDB ID: 9P9P, 1.48 Å), 5 (PDB ID: 9PAU, 1.7 Å), 9 (PDB ID: 9PBN, 1.46 Å), 10 (PDB ID: 9PBU, 1.73 Å), 13 (PDB ID: 9PCL, 1.76 Å), and 15 (PDB ID: 9PCM, 1.45 Å), the reported R-values appear relatively high (R_{work}/R_{free} [structure 3] = 0.20/0.22; R_{work}/R_{free} [structure 5] = 0.21/0.24; R_{work}/R_{free} [structure 9] = 0.20/0.22; R_{work}/R_{free} [structure 10] = 0.20/0.23; R_{work}/R_{free} [structure 13] = 0.20/0.24; R_{work}/R_{free} [structure 15] = 0.20/0.22) considering the resolution of the datasets. The reviewer recommends that the authors re-examine the model coordinates and consider reprocessing the diffraction data using an appropriate resolution cutoff to improve model quality and ensure coordinate reliability.
2. The authors should clearly state whether the substrate 3-(carboxymethyl)amino decanoic acid (CADA) used in this study is a single isomer, e.g., C2-R, or a racemic mixture at the beta-position. The C3-chirality of CADA is not assigned in either Scheme 1 or the Methods section. According to references 12 and 44 cited in the Methods, the starting material for CADA consists of a mixture of R and S isomers rather than a single stereoisomer. If a racemic mixture was indeed used, this raises concerns regarding the accuracy and interpretation of the reported kinetic parameters for CADA.
3. To validate the importance of selected residues in substrate binding and specificity, the authors conducted enzyme activity assays using CADA and determined kinetic parameters such as k_{cat} and K_M. In most cases, the variants exhibited reduced activity toward CADA. To gain insight into these variants, the authors should consider profiling the variants against a series of substrates, e.g., those vary alkyl chain lengths. It is possible that the variants retain catalytic activity but alter the substrate preferences compared to the wild type. A similar approach has been demonstrated in a recent publication (*RSC Chem. Biol.* 2025, 6, 583–589).
4. The label of the y-axes in Figures 2H, 3I, and 4B is unclear and should be revised for better readability and interpretation. Additionally, there appear to be errors in the variant names shown on the x-axes of these figures. The authors should carefully check and correct the labeling to ensure consistency with the text and figure legends.
5. The authors propose that, following the first half-reaction, the imine intermediate remains bound within the active site rather than undergoing dissociation and re-binding process. This is an important statement, but currently lacking experimental support. Moreover, the kinetic data cited from references 11 and 14 raise questions about the proposed model. Specifically, the reported K_M value for the imine intermediate derived from CABA (K_M = 369 ± 40 μM) is higher than that of the parent substrate CABA (K_M = 286 ± 93 μM). This suggests that the substrate binds slightly tighter than the intermediate, which does not support the proposed model. The authors should address this inconsistency, clarify how these kinetic values

align with their proposed mechanism, and provide additional experimental evidence to support their hypothesis.

6. The authors discuss the concepts of online and offline 2OG binding, along with potential conformational changes, by comparing their ternary structure (structure 15) with the Fe(IV)-oxo mimic vanadyl-bound structures. However, no side-by-side structural comparison figures are provided in the main text or SI. Without such comparisons, it is difficult for readers to evaluate the proposed differences in binding modes and associated conformational shifts. The authors are encouraged to include comparative figures to support and clarify their discussion.

Reviewer #2

(Remarks to the Author)

The authors report 16 structures of wild-type or selected variants of Rv0097, an intriguing isonitrile biosynthetic enzyme from *Mycobacterium tuberculosis*. Significantly, the structures include those of the active Fe(II)-containing enzyme bound to alpha-ketoglutarate (aKG), to 3-carboxymethylaminodecanoic acid (CADA), and to both substrates. The results are consistent with a reasonable two-step enzyme mechanism that includes an unstable imine intermediate, where each step requires oxidative cleavage of aKG and involves conformational changes of several residues on two loops. The structures also explain the preference of this enzyme for substrates possessing long alkyl groups in comparison to the preference for short alkyl groups by ScoE, a related isonitrile-forming enzyme that was previously characterized structurally. The current structural studies are nicely complemented by activity measurements of selected variants. The work is of high quality and should be of great interest to many readers. The writing is logically presented and very clear, although numerous minor concerns are present as indicated below:

Line 47: Figure S1A shows the Trp nitrogen atom in red changing to blue in the product with the product carbon atom in red from an unclear carbon of Ru5P. An alternative coloration scheme could help a reader more clearly see the sources of the N and C atoms.

Line 56: "These ... synthesis" has inconsistent number agreement.

Line 67: Figure S1B confusingly shows the N and C changing color during the reaction.

Line 71: "ScoE/Rv0097 is" has inconsistent number agreement and would be better written as "ScoE and Rv0097 are".

Line 82: Figure S3 shows the C4 methyl group of butyrate transforming into a hydroxyl group rather than the glyceryl moiety undergoing hydroxylation. Two of the structures in this figure are incorrect.

Line 107: "apo" is lab jargon and should be defined at first use as the apoprotein.

Line 108: It would be helpful to include the PDB ID for each structure in Table S1 despite Table S2 summarizing this information. State the method for calculating Rmerge in Table S1.

Line 114: Delete "two" for clarity since there is only one His and one Arg of interest.

Line 120: The legend of Figure S5 should merge two sentence fragments ("When modeled as a Mg ion. No mFo-DFc difference...") into a single sentence in line 3 and insert the word "be" into "to consistent" in line 4.

Line 134: To contrast with the bulky F239 of ScoE It might help the reader see the increased space if G204 of Rv97 is included in Figure S6.

Line 145: Perhaps specify the residues involved in the hydrophobic interactions with CADA in Figure 1E and clarify if they are conserved in ScoE.

Line 148: Only DNA, not protein, is mutated so this line should be rewritten as "mutation of the codon for G204 to encode" or "substitution of G204".

Line 166: Can the authors provide an estimate for the occupancy of CABA?

Line 168: Consider indicating that the data in panels A and B were obtained from triplicate measurements as mentioned in Methods? These are rate versus substrate concentration plots, not Michaelis-Menten kinetics plots (the plots were fit to Michaelis-Menton kinetics). The table sometimes includes an unrealistic number of significant digits (e.g. 8.07 +/- 1.57) and in some cases the number of significant digits in the value does not agree with the number of significant digits in the error (e.g. 1.72 +/- 0.4).

Line 170: Perhaps define (as wild-type enzyme) this first use of "WT".

Line 176: The sentence is missing "of" after "roles".

Line 178: The sentence is misleading because CADA was not bound well to the F102A variant as implied for structure 11. Perhaps this sentence should indicate that the complexes with CADA also included vanadate. Perhaps Table S2 should indicate CADA was included for structure 11, even though it bound poorly.

Line 222: The legend to Figure S9 should indicate these represent triplicate results and replace the phrase "Michaelis-Menten kinetics plot".

Line 227: Perhaps also state the substrate concentration so a reader will know if the enzyme kinetics studies used saturating conditions?

Line 243: CADA soaking appears to not only crack or deform but also enlarge the crystals in Figure S10. Is this the case?

Lines 257 and 258: "half ... undergoes", not "half ... undergo".

Line 278: Perhaps state the RMSD between the one open chain and the three closed states for the 10 mM CADA-soaked sample.

Line 285: ScoE R157 is said to be equivalent to R122 of Rv0097 and several other comparable residues in these proteins are mentioned elsewhere. The authors should consider including a sequence alignment of the two proteins in the supplementary material.

Lines 287 and 302: The terms offline and online when referring to aKG conformation should be defined (perhaps include a diagram?) and cite a reference.

Line 344: The stated 1.45-Å resolution for structure 16 differs from what is shown in Table S1 of 2.5-Å.

Lines 418-420: This was the first time the manuscript mentions the previously published structural analysis of Rv0097. The text should state the access code for that structure (8KHT), indicate it was for the complex of holoenzyme with substrate, and

compare the prior results to the holoenzyme + CADA structure reported here, perhaps where structure 2 is described.
Line 477: insert "reactive" into "highly iron-oxygen intermediate".
Line 532: Table S5 should include the sources of the strains.
Line 535: Define TB?
Line 541: Define NTA?
Line 556: WT should have been defined much earlier and can be used here without definition.
Line 557: Consider changing "Reconstruction" to "Metallation".
Line 563-564: CADA and CABA were defined earlier and can be used without doing so again. Perhaps later in this paragraph the definitions could be provided for HRMS, NMR, HPLC?
Line 595: Py-tetrazine was just defined in line 580; no need to duplicate.
Line 586: INDA was already defined and can be used without redefinition.
Lines 612 and 614: Replace "media" with "medium".
Lines 620 and 628: The g should be italicized.
Line 635: EDTA was used earlier where the definition should be found so the definition is not needed here.
Line 648: Perhaps state the initial crystallization screens?
Line 671: Redefining aKG is not needed here. Also, portions of this and the following two paragraphs (e.g., "harvested, cryoprotected with paraffin oil, and flash frozen in liquid nitrogen") are redundant; can the descriptions be simplified?
Lines 725, 735, 749, 760, and 777: The authors used 2|Fo|-|Fc| maps contoured to 1.0 σ for manual adjustment during model building. Why was this chosen instead of the more typical 2mFo-DFc maps?
Lines 726 and later: What were the contour levels of the 2mFo-DFc maps?
Line 729 and later: Please state how the |Fo|-|Fc| maps at 3.0 σ were generated.
Line 732: Was molecular replacement used for phasing structures 2-7, 9,15 and 16?
Line 975: Consider showing the physiological protonation states for parts A and B. More importantly, the oxidation state of iron in the hydroxyl intermediate is an issue. The hydroxylation of the glycyl group would result in an Fe(II) state of the enzyme. Is there evidence for coordination of the alkoxide to the metal ion?
Line 982: Label D99 in Figure 1B.
Line 985: Fe should be replaced by Mg.
Line 986: It is mentioned on line 131, but it might be useful to restate here the occupancies of the two F102 rotamers.
Line 987: Perhaps replace "only one" with "each single"?
Line 995: Perhaps repeat the information from line 165 on the partial occupancy of CABA in panel H.
Line 996: Please mention the map type and contour level for panels G, H and I.
Line 998: The organization of the panels would be more logical by shifting panel B into the middle of the top row. The table in G has issues with the numbers of significant figures and in one case the agreement between the value and the error. The y-axis label and values and the labels in the x-axis have issues in panel H. Also, the triplicate points should be shown, not just error bars.
Line 1001: F102 should read F102W or W102.
Lines 1003-1009: Indicate the map type and contour level for panels 2B, C, D & E.
Line 1011: Perhaps indicate the assay had used 1 mM substrate (far from saturating for F102A!) and was carried out for 10 min?
Line 1014. The upper two number values on the Panel I y-axis and the sample labels (PA53A and PA53V) need attention. Showing the triplicate points would be appropriate.
Line 1027: The centroid-to-centroid distance in Figure 3H is for the F102', not F102, conformation in the open state.
Line 1031: Panel B has errors in the y-axis title and labels as well as the names of the variants studied. Showing the triplicate points would be a nice addition.

Reviewer #3

(Remarks to the Author)

Version 1:

Reviewer comments:

Reviewer #1

(Remarks to the Author)

This reviewer appreciates the authors' detailed clarification and recognizes their extensive expertise in structural biology and crystallization. The explanation addresses the concern regarding potential lattice artifacts and highlights the exceptional crystallographic behavior of Rv0097. However, the biochemical evidence obtained using the racemic substrate, racemic-CADA, remains unsolved and raises several concerns that could impact the conclusion drawn in this manuscript.

Enantiomerically pure CADA has been synthesized on a gram scale in several reported literature including PNAS, 2022, 119, e2110293119; ACS Catal. 2024, 14, 4975–4983; J. Am. Chem. Soc. 2025 DOI: 10.1021/jacs.5c13262. Therefore, the use of a racemic compound should not be considered as a technical limitation. Unless the authors clearly establish that the S-isomer does not influence the kinetic measurements or the interpretation of intermediate retention within the active site, the

current data they present seem unclear to support the conclusion. Specifically, it should be clarified whether the S-isomer is a competent substrate for Rv0097, and if so, whether it proceeds to the isonitrile product or instead lead to hydroxylated or imine intermediate formation. Furthermore, given that authors emphasize the role of Rv0097 variants in governing substrate binding and specificity, it is important to address whether these variants can affect the binding affinity or reactivity of the S-isomer. Moreover, the authors should acknowledge relevant contributions from other researchers and incorporate the references in the main text, rather than citing them only in the response letter. To merit publication in a high-impact journal, the authors are encouraged to clearly address the issues outlined above.

Reviewer #2

(Remarks to the Author)

All of my prior concerns have been adequately addressed.

One minor point to fix is in the legend to Figure S5; the word "density" is crossed out but should be deleted.

Reviewer #3

(Remarks to the Author)

Reviewer #1 (Remarks to the Author):

The authors reported several high-resolution crystal structures of the isonitrile biosynthetic enzyme Rv0097 from *Mycobacterium tuberculosis*, including both wild-type and variant forms in the presence and absence of substrate (CADA) and α -ketoglutarate (α KG). These structural insights help elucidate the enzyme's architecture and suggest potential conformational changes that may contribute to isonitrile formation.

Thank you very much for your comments.

However, the current manuscript lacks experimental evidence to substantiate the conclusions drawn from these structural snapshots. To this reviewer, it remains unclear whether the observed conformation differences correspond to catalytically relevant intermediates or they are simply artifacts during crystallization.

The reviewer is correct that crystal lattices can lead to artifacts. All too often the crystal lattice prevents movements that are relevant to catalysis. However, Rv0097 is not a protein system that suffers from this issue. We apologize if our writing of this manuscript did not make this clear. The reason that this paper has 16 structures is that Rv0097 crystallizes with lattices that are malleable to conformational change, and when the lattice cannot accommodate the required conformational change, Rv0097 crystallized in different space groups. My lab has deposited over 200 entries to the Protein Data Bank, and I have never before worked on a protein system like Rv0097 that was so amenable to crystallization in so many different conformational states. We were incredibly fortunate in this regard. Additionally, our proposed mechanism based on these structural snapshots are further supported by biochemical studies and are in line with previous proposed mechanisms of related non-heme Fe(II)- α KG-dependent dioxygenase. We have edited the text to explain these points clearly.

In addition, several statements in the manuscript overlap with the authors' previous work and other published literatures.

Thank you for your comment. Our prior work on Rv0097 (*Biochemistry* 2023, 62, 3, 824–834) identified G204 residue as a putative residue gating substrate specificity. Later work by others (*ACS Catal.* 2024, 14, 4975–4983 and *RSC Chem. Biol.* 2025, 6, 583–589) reported preliminary structural data identifying residues in the substrate-binding pocket that govern substrate alkyl chain length specificity by differential side chain bulkiness. In this work, we revealed an additional layer of specificity control of Rv0097 by a pair of substrate-binding site residues, F102/G204, that gate substrate binding and specificity by maintaining a hydrophobic environment. We also report conformational changes of other regions of the enzyme that were not previously observed. This study showcases how multiple flexible regions achieve two half reactions in one catalytic cycle in a highly orchestrated fashion and highlights the role of enzyme dynamics in controlling activities. We are building off of previous work. We have edited the text to make sure that it is clear as to what was known and what is newly discovered.

To merit publication in a high-impact journal such as *Nature Communications*, the authors should consider carrying out experiments to provide insights into the proposed mechanism. Specific comments are listed below:

1. For structures 3 (PDB ID: 9P9P, 1.48 Å), 5 (PDB ID: 9PAU, 1.7 Å), 9 (PDB ID: 9PBN, 1.46 Å), 10 (PDB ID: 9PBU, 1.73 Å), 13 (PDB ID: 9PCL, 1.76 Å), and 15 (PDB ID: 9PCM, 1.45 Å), the reported R-values appear relatively high ($R_{\text{work}}/R_{\text{free}}$ [structure 3] = 0.20/0.22; $R_{\text{work}}/R_{\text{free}}$ [structure 5] = 0.21/0.24; $R_{\text{work}}/R_{\text{free}}$ [structure 9] = 0.20/0.22; $R_{\text{work}}/R_{\text{free}}$ [structure 10] = 0.20/0.23; $R_{\text{work}}/R_{\text{free}}$ [structure 13] = 0.20/0.24; $R_{\text{work}}/R_{\text{free}}$ [structure 15] = 0.20/0.22) considering the resolution of the datasets. The reviewer recommends that the authors re-examine the model coordinates and consider reprocessing the diffraction data using an appropriate resolution cutoff to improve model quality and ensure coordinate reliability.

All our atomic models were carefully examined residue-by-residue after every round of structure refinement against $2mF_o-DF_c$ omit maps.

Although truncating the data to lower resolution will lower R-factors, throwing away higher resolution data cannot be justified in a case where the R-factors are appropriate for structures that are not over-refined. The gaps between R_{work} and R_{free} are very small, indicating proper structure refinements with no overfitting. Our current resolution cutoffs were determined based on holistic considerations of statistics during data processing: R_{sym} , $CC_{1/2}$, I/σ , and data completeness.

Additionally, R-factors lower than what we report here would not be expected for a small protein like Rv0097 (with molecular weight of ~30 kDa) that has multiple regions that are flexible. Although we worked hard to model the flexible regions, using alternative conformations as appropriate, it is not possible to build a structural model that accounts for all of the protein's flexibility. Of course, all proteins have at least one flexible loop and thus the agreement between F_{obs} and F_{calc} is never perfect, but when a large percentage of a protein is flexible as is the case here, R-factors close to 20% is what one would expect for structures that are not overrefined.

Although we were satisfied with our R-factors, we did subject our structures to further refinement to address this reviewers' concern. We performed an additional final round of Translation-Libration-Screw (TLS) refinement for each of the above-mentioned structures. The R-factors are lowered for some structures (structures **3**, **9**, and **10**) whereas they stayed roughly the same for the others. The table below summarizes these changes in R-factors, and the new R-factors are now added to **Table S1**. We examined the maps individually before and after TLS refinements and found *no noticeable differences*. Please also see screenshots below.

R-factors before and after a final round of TLS refinement			
Structure	Resolution (Å)	Before TLS refinement (current) R_{work}/R_{free}	After TLS refinement R_{work}/R_{free}
3	1.48	0.199/0.219	0.179/0.195
5	1.70	0.211/0.236	0.205/0.232
9	1.46	0.195/0.218	0.175/0.192
10	1.73	0.195/0.228	0.177/0.210
13	1.76	0.198/0.237	0.194/0.230
15	1.45	0.195/0.218	0.193/0.215

The following images are screenshots of $2mF_o-DF_c$ maps (1.5σ , blue mesh) and mF_o-DF_c maps ($+3.0 \sigma$, green mesh, -3.0σ , red mesh) with atomic models of structures **3**, **9**, and **10** fit in before and after one final round of TLS refinement:

2. The authors should clearly state whether the substrate 3-(carboxymethyl)amino decanoic acid (CADA) used in this study is a single isomer, e.g., C2-R, or a racemic mixture at the beta-position. The C3-chirality of CADA is not assigned in either Scheme 1 or the Methods section. According to references 12 and 44 cited in the Methods, the starting material for CADA consists of a mixture of R and S isomers rather than a single stereoisomer. If a racemic mixture was indeed used, this raises concerns regarding the accuracy and interpretation of the reported kinetic parameters for CADA.

In this study, we utilized a racemic mixture of CADA (at the β -position) for the biochemical characterization of Rv0097, as detailed in our previous publication in *Biochemistry* (DOI: 10.1021/acs.biochem.2c00611). The current version of the manuscript has been revised to explicitly clarify this point.

Regarding the use of a racemic mixture of CADA, we did consider synthesizing the enantiomerically pure *R*-isomer of CADA. However, this approach would require a significant investment of time (estimated at over 2–3 months) and incur substantial material costs (exceeding \$3,000 with an estimated lead time of 30 days). While we recognize that using enantiomerically pure CADA would be ideal, we are confident in the validity of our kinetic parameters. All enzyme variants were assessed

using the same *R/S* mixture of CADA, thus allowing for reliable comparative analysis across the different enzyme variants.

3. To validate the importance of selected residues in substrate binding and specificity, the authors conducted enzyme activity assays using CADA and determined kinetic parameters such as k_{cat} and K_M . In most cases, the variants exhibited reduced activity toward CADA. To gain insight into these variants, the authors should consider profiling the variants against a series of substrates, e.g., those vary alkyl chain lengths. It is possible that the variants retain catalytic activity but alter the substrate preferences compared to the wild type. A similar approach has been demonstrated in a recent publication (RSC Chem. Biol. 2025, 6, 583–589).

We agree with the reviewer that profiling the variants against different substrates is important to probe residues potentially affecting chain length recognition, which has been extensively studied by us, by comparing CADA vs. CABA (this manuscript and *Biochemistry* 2023, 62, 824-834), and other researchers (*ACS Catal.* 2024, 14, 4975-4983; *RSC Chem. Biol.* 2025, 6, 583–589). However, we feel that our kinetic analysis toward CADA is sufficient to probe the role of the F102/G204 pair in governing substrate binding and specificity, the role of P153 in gating CADA binding, and the role of H264/R122 in gating binding of co-substrates.

4. The label of the y-axes in Figures 2H, 3I, and 4B is unclear and should be revised for better readability and interpretation. Additionally, there appear to be errors in the variant names shown on the x-axes of these figures. The authors should carefully check and correct the labeling to ensure consistency with the text and figure legends.

Thank you for pointing this issue out. The problem arose in the PDF conversion that was performed on the journal website. I am not sure how to fix this problem since the issue does not arise when the PDF is viewed on a home computer. We remade these figure panels with the hope of improved compatibility and will check the conversion before submission.

5. The authors propose that, following the first half-reaction, the imine intermediate remains bound within the active site rather than undergoing dissociation and re-binding process. This is an important statement, but currently lacking experimental support. Moreover, the kinetic data cited from references 11 and 14 raise questions about the proposed model. Specifically, the reported K_M value for the imine intermediate derived from CABA ($K_M = 369 \pm 40 \mu\text{M}$) is higher than that of the parent substrate CABA ($K_M = 286 \pm 93 \mu\text{M}$). This suggests that the substrate binds slightly tighter than the intermediate, which does not support the proposed model. The authors should address this inconsistency, clarify how these kinetic values align with their proposed mechanism, and provide additional experimental evidence to support their hypothesis.

To investigate whether the imine-CADA intermediate remains sequestered within the enzyme active site, we have performed an additional series of biochemical assays employing a previously established method for the indirect detection of the imine via chemical derivatization of glyoxylate under acidic conditions (DOI: 10.1021/jacs.1c12891). Reactions were quenched after short incubation periods (1 minute), followed by chemical derivatization to assess glyoxylate formation for imine-CADA intermediate quantification. To distinguish between free and enzyme-bound imine-CADA, we compared the relative amounts in the reaction supernatant alone (enzyme is removed via filtration) to that of the complete reaction mixture containing the enzyme (see new SI Figure, **Fig. S20**). The results from this new experiment show substantially more imine-CADA in the sample with the enzyme present, consistent with imine-CADA remaining bound in the active site during catalysis, and consistent with imine-CADA having a relatively high affinity for the enzyme. The somewhat higher K_M for the imine-CADA compared to CADA does not conflict with our proposal that the imine-CADA remains bound to the enzyme during catalysis. A somewhat higher K_M merely indicates that imine-CADA is a slightly poorer substrate than CADA if added exogenously. The new data inform on binding affinity of imine-CADA rather than on its K_M and are a nice addition to the paper. We thank the reviewer for encouraging us to perform additional experiments.

We have included this new experiment in the Results section of the main text and have referenced the new SI figure (Fig. S20).

6. The authors discuss the concepts of online and offline 2OG binding, along with potential conformational changes, by comparing their ternary structure (structure 15) with the Fe(IV)-oxo mimic vanadyl-bound structures. However, no side-by-side structural comparison figures are provided in the main text or SI. Without such comparisons, it is difficult for readers to evaluate the proposed differences in binding modes and associated conformational shifts. The authors are encouraged to include comparative figures to support and clarify their discussion.

Thank you for your suggestion. We have now added a new SI figure (Fig. S16) that provides these side-by-side comparisons.

Reviewer #2 (Remarks to the Author):

The authors report 16 structures of wild-type or selected variants of Rv0097, an intriguing isonitrile biosynthetic enzyme from Mycobacterium tuberculosis. Significantly, the structures include those of the active Fe(II)-containing enzyme bound to alpha-ketoglutarate (aKG), to 3-carboxymethylaminodecanoic acid (CADA), and to both substrates. The results are consistent with a reasonable two-step enzyme mechanism that includes an unstable imine intermediate, where each step requires oxidative cleavage of aKG and involves conformational changes of several residues on two loops. The structures also explain the preference of this enzyme for substrates possessing long alkyl groups in comparison to the preference for short alkyl groups by ScoE, a related isonitrile-forming enzyme that was previously characterized structurally. The current structural studies are nicely complemented by activity measurements of selected variants. The work is of high quality and should be of great interest to many readers. The writing is logically presented and very clear, although numerous minor concerns are present as indicated below.

Thank you very much for your very detailed comments and suggestions that helped us greatly to improve the quality of our manuscript.

Line 47: Figure S1A shows the Trp nitrogen atom in red changing to blue in the product with the product carbon atom in red from an unclear carbon of Ru5P. An alternative coloration scheme could help a reader more clearly see the sources of the N and C atoms.

Corrected.

Line 56: "These ... synthesis" has inconsistent number agreement.

Corrected.

Line 67: Figure S1B confusingly shows the N and C changing color during the reaction.

Corrected.

Line 71: "ScoE/Rv0097 is" has inconsistent number agreement and would be better written as "ScoE and Rv0097 are".

Changed to "ScoE and its homologs are..."

Line 82: Figure S3 shows the C4 methyl group of butyrate transforming into a hydroxyl group rather than the glycyl moiety undergoing hydroxylation. Two of the structures in this figure are incorrect.

Corrected. Thank you.

Line 107: "apo" is lab jargon and should be defined at first use as the apoprotein.

“Apo” is now first defined as apoprotein.

Line 108: It would be helpful to include the PDB ID for each structure in Table S1 despite Table S2 summarizing this information. State the method for calculating Rmerge in Table S1.

PDB IDs for each structure are now added to Table S1. Per standard formatting guidelines for Nature journals, equations defining various R-values are standard and hence are no longer defined in the footnotes (<https://www.nature.com/nature/for-authors/formatting-guide>).

Line 114: Delete “two” for clarity since there is only one His and one Arg of interest.

Corrected.

Line 120: The legend of Figure S5 should merge two sentence fragments (“When modeled as a Mg ion. No mFo-DFc difference...”) into a single sentence in line 3 and insert the word “be” into “to consistent” in line 4.

Corrected.

Line 134: To contrast with the bulky F239 of ScoE It might help the reader see the increased space if G204 of Rv97 is included in Figure S6.

Fig. S6 is now updated to include G204 (α carbon shown as a sphere) of Rv0097.

Line 145: Perhaps specify the residues involved in the hydrophobic interactions with CADA in Figure 1E and clarify if they are conserved in ScoE.

We added a list of residues involved in hydrophobic interactions with CADA. We also added a sequence alignments SI figure (**Fig. S7**) highlighting these residues across Rv0097 homologs: ScoE and MmaE. We additionally added an SI figure (**Fig. S9**) showing that two loops in the substrate-binding pocket are positioned differently in ScoE and Rv0097 to accommodate the different substrate alkyl chain lengths.

Line 148: Only DNA, not protein, is mutated so this line should be rewritten as “mutation of the codon for G204 to encode” or “substitution of G204”.

Corrected.

Line 166: Can the authors provide an estimate for the occupancy of CABA?

Yes. Refined occupancy of CABA is 0.65 in this structure. We have added this information to the text.

Line 168: Consider indicating that the data in panels A and B were obtained from triplicate measurements as mentioned in Methods? These are rate versus substrate concentration plots, not Michaelis-Menten kinetics plots (the plots were fit to Michaelis-Menton kinetics). The table sometimes includes an unrealistic number of significant digits (e.g. 8.07 +/- 1.57) and in some cases the number of significant digits in the value does not agree with the number of significant digits in the error (e.g. 1.72 +/- 0.4).

We have added in the figure legend of **Fig. S10** (old **Fig. S8**) that data in panels A and B were obtained from triplicate measurements. We have edited the legends to be velocity vs. substrate concentration plots. The tables now include values with correct numbers of significant digits, and that the number of significant digits of error are consistent with the number of significant digits of the values. We also made sure that errors of k_{cat}/K_M are calculated following the rules for propagation of uncertainty.

Line 170: Perhaps define (as wild-type enzyme) this first use of “WT”.

WT is now first defined as wild-type enzyme.

Line 176: The sentence is missing “of” after “roles”.

Corrected.

Line 178: The sentence is misleading because CADA was not bound well to the F102A variant as implied for structure 11. Perhaps this sentence should indicate that the complexes with CADA also included vanadate. Perhaps Table S2 should indicate CADA was included for structure 11, even though it bound poorly.

We edited the text and Table S2 as suggested.

Line 222: The legend to Figure S9 should indicate these represent triplicate results and replace the phrase “Michaelis-Menten kinetics plot”.

We have added in the figure legend of **Fig. S11** (old **Fig. S9**) that data in panels A and B were obtained from triplicate measurements. We have edited the legends to be velocity vs. substrate concentration plots.

Line 227: Perhaps also state the substrate concentration so a reader will know if the enzyme kinetics studies used saturating conditions?

Substrate concentration for assay (1 mM) is added. This is not saturating concentration – we use the term “saturating” here per definition of k_{cat} as we discuss and compare k_{cat} values of different enzyme variants.

Line 243: CADA soaking appears to not only crack or deform but also enlarge the crystals in Figure S10. Is this the case?

Crystals are not enlarged. Sorry, the photos are misleading in this regard. Photos of crystals before and after the soak are zoomed-in differently. Deformed crystals are more zoomed in to show the cracks more clearly. We have added this information in the figure legend.

Lines 257 and 258: “half ... undergoes”, not “half ... undergo”.

Corrected.

Line 278: Perhaps state the RMSD between the one open chain and the three closed states for the 10 mM CADA-soaked sample.

We have added RMSD values in the paragraph describing the overlaid open and closed structures.

Line 285: ScoE R157 is said to be equivalent to R122 of Rv0097 and several other comparable residues in these proteins are mentioned elsewhere. The authors should consider including a sequence alignment of the two proteins in the supplementary material.

We added a new SI figure (**Fig. S7**) providing sequence alignment of Rv0097, ScoE, and MmaE, highlighting these residues.

Lines 287 and 302: The terms offline and online when referring to aKG conformation should be defined (perhaps include a diagram?) and cite a reference.

Definitions of offline have been added to the text. We included a new SI figure (**Fig. S16**) to show online and offline α KG/Fe(IV)-oxo conformations.

Line 344: The stated 1.45-Å resolution for structure 16 dimers from what is shown in Table S1 of 2.5-Å.

The resolution of structure 16 is 1.36 Å (not 1.45 Å). We have fixed this value in the text. It was correct in Table S1. Thank you for noticing this typo. The resolution of the anomalous data for this structure was 2.5 Å.

Lines 418-420: This was the first time the manuscript mentions the previously published structural analysis of Rv0097. The text should state the access code for that structure (8KHT), indicate it was for

the complex of holoenzyme with substrate, and compare the prior results to the holoenzyme + CADA structure reported here, perhaps where structure 2 is described.

We now reference this structure in the introduction. Additionally, the PDB accession code 8KHT is also added to lines 418-420. A structural similarity comparison is added to the location in the text where structure 2 is first mentioned.

Line 477: insert “reactive” into “highly iron-oxygen intermediate”.

Corrected.

Line 532: Table S5 should include the sources of the strains.

Sources of strains have been added.

Line 535: Define TB?

TB defined.

Line 541: Define NTA?

NTA defined.

Line 556: WT should have been defined much earlier and can be used here without definition.

WT is now defined earlier and the definition here is removed.

Line 557: Consider changing “Reconstruction” to “Metallation”.

We changed “Reconstruction” to “Reconstitution” to be consistent with terms we used in our previous works.

Line 563-564: CADA and CABA were defined earlier and can be used without doing so again. Perhaps later in this paragraph the definitions could be provided for HRMS, NMR, HPLC?

Definitions of CADA and CABA here are removed. HRMS, NMR, HPLC are defined.

Line 595: Py-tetrazine was just defined in line 580; no need to duplicate.

Duplicate definition is removed.

Line 586: INDA was already defined and can be used without redefinition.

We removed “isonitrile” and now the sentence reads “The production of INDA...”

Lines 612 and 614: Replace “media” with “medium”.

Corrected.

Lines 620 and 628: The g should be italicized.

Corrected.

Line 635: EDTA was used earlier where the definition should be found so the definition is not needed here.

EDTA is now first defined in the first paragraph of Results section and the following duplicate definitions are removed.

Line 648: Perhaps state the initial crystallization screens?

Initial crystallization screen added.

Line 671: Redefining α KG is not needed here. Also, portions of this and the following two paragraphs (e.g., “harvested, cryoprotected with paraffin oil, and flash frozen in liquid nitrogen”) are redundant; can the descriptions be simplified?

α KG is defined here to indicate the nature of the salt and to give source of commercially purchased α KG used. We have trimmed the redundancy throughout the methods.

Lines 725, 735, 749, 760, and 777: The authors used $2|F_o|-|F_c|$ maps contoured to 1.0σ for manual adjustment during model building. Why was this chosen instead of the more typical $2mF_o-DF_c$ maps?

The maps we used were indeed σ_A -weighted $2mF_o-DF_c$ and mF_o-DF_c maps for model building in Coot (outputs of phenix.refine). We now specifically indicate that the maps were weighted.

Lines 726 and later: What were the contour levels of the $2mF_o-DF_c$ maps?

We have double checked that contour levels are provided.

Line 729 and later: Please state how the $|F_o|-|F_c|$ maps at 3.0σ were generated.

All $2mF_o-DF_c$ and mF_o-DF_c maps are generated by phenix.refine. This information is now added.

Line 732: Was molecular replacement used for phasing structures 2-7, 9,15 and 16?

No. When the space group was the same, molecular replacement was not needed. We started with rigid-body refinement using the apo structure 1. This information is already included in the text.

Line 975: Consider showing the physiological protonation states for parts A and B. More importantly, the oxidation state of iron in the hydroxyl intermediate is an issue. The hydroxylation of the glycyI group would result in an Fe(II) state of the enzyme. Is there evidence for coordination of the alkoxide to the metal ion?

We updated Scheme 1A and 1B. with relevant protonation states of molecules. We do not have evidence for alkoxide coordination to the iron. We replaced the alkoxide species with the hydroxylated intermediate and corrected the oxidation state of iron to Fe(II).

Line 982: Label D99 in Figure 1B.

D99 labeled.

Line 985: Fe should be replaced by Mg.

Corrected.

Line 986: It is mentioned on line 131, but it might be useful to restate here the occupancies the two F102 rotamers.

F102 rotamer occupancies added.

Line 987: Perhaps replace “only one” with “each single”?

We changed to "a single".

Line 995: Perhaps repeat the information from line 165 on the partial occupancy of CABA in panel H.

Added.

Line 996: Please mention the map type and contour level for panels G, H and I.

Map information included.

Line 998: The organization of the panels would be more logical by shifting panel B into the middle of the top row. The table in G has issues with the numbers of significant figures and in one case the

agreement between the value and the error. The y-axis label and values and the labels in the x-axis have issues in panel H. Also, the triplicate points should be shown, not just error bars.

We have reorganized panels in **Fig. 2** so that each row shows one variant. Significant figures in **Fig. 2G** have been fixed. Some of the issues with Fig. 2H have to do with the journal's PDF converter and are out of our control, but we did add individual data points in addition to the error bars.

Line 1001: F102 should read F102W or W102.

Corrected.

Lines 1003-1009: Indicate the map type and contour level for panels 2B, C, D & E.

We were trying to avoid redundancy but now repeat the map information in each panel description.

Line 1011: Perhaps indicate the assay had used 1 mM substrate (far from saturating for F102A!) and was carried out for 10 min?

We have added this information.

Line 1014. The upper two number values on the Panel I y-axis and the sample labels (PA53A and PA53V) need attention. Showing the triplicate points would be appropriate.

Most of the issues with Figure 3I were caused by the Journal's PDF converter, but we have added individual data points in addition to the error bars. We also remade the panel in the hope of improving compatibility with the PDF converter.

Line 1027: The centroid-to-centroid distance in Figure 3H is for the F102', not F102, conformation in the open state.

Corrected.

Line 1031: Panel B has errors in the y-axis title and labels as well as the names of the variants studied. Showing the triplicate points would be a nice addition.

Most of the issues with Figure 4B were caused by the Journal's PDF converter, but we have added individual data points in addition to the error bars. We also remade the panel in the hope of improving compatibility with the PDF converter.

Reviewer #3 (Remarks to the Author):

Thank you for assisting in this review.

POINT BY POINT

Reviewer #1 (Remarks to the Author):

This reviewer appreciates the authors' detailed clarification and recognizes their extensive expertise in structural biology and crystallization. The explanation addresses the concern regarding potential lattice artifacts and highlights the exceptional crystallographic behavior of Rv0097. However, the biochemical evidence obtained using the racemic substrate, racemic-CADA, remains unsolved and raises several concerns that could impact the conclusion drawn in this manuscript.

Thank you.

Enantiomerically pure CADA has been synthesized on a gram scale in several reported literature including PNAS, 2022, 119, e2110293119; ACS Catal. 2024, 14, 4975–4983; J. Am. Chem. Soc. 2025 DOI: 10.1021/jacs.5c13262. Therefore, the use of a racemic compound should not be considered as a technical limitation. Unless the authors clearly establish that the S-isomer does not influence the kinetic measurements or the interpretation of intermediate retention within the active site, the current data they present seem unclear to support the conclusion. Specifically, it should be clarified whether the S-isomer is a competent substrate for Rv0097, and if so, whether it proceeds to the isonitrile product or instead lead to hydroxylated or imine intermediate formation. Furthermore, given that authors emphasize the role of Rv0097 variants in governing substrate binding and specificity, it is important to address whether these variants can affect the binding affinity or reactivity of the S-isomer.

Previous work (ACS Catal. 2024, 14, 4975–4983, ref. 26) showed that the S-isomer of CADA has negligible activity for isonitrile formation by Rv0097 compared to the R-isomer. The authors observed over 20-fold decrease in enzymatic activity when S-isomer is used as the substrate. We have now provided additional statements in the Results section that explicitly reference this work. In addition, we clearly observe R-CADA binding in the active site in all our CADA-bound WT and variant crystal structures despite the use of the racemic mixture, suggesting weak competition from S-CADA. We mention this fact explicitly in the Results:

Lines 138–141: We note here that we used racemic-CADA for co-crystallization due to substantial costs and time investments it requires to synthesize the enantiopure (R)-CADA²⁶. We find only the (R) isomer bound in the structure, indicating the preference of the enzyme for (R)-CADA over (S)-CADA, as has been previously reported²⁶

We also now acknowledge in the Results that the use of a racemic mixture of CADA is a caveat that should be keep in mind:

Lines 176–180: As with the structural studies, we used a racemic mixture of CADA in all our assays. Since we don't see evidence of (S)-CADA binding in our structures, we believe that the impact of the racemic CADA mixture on enzyme activity is minor, consistent a with previous report²⁶. However, the use of a racemic mixture of CADA and enantiomeric pure CADA is a caveat that should be kept in mind.

Moreover, the authors should acknowledge relevant contributions from other researchers and incorporate the references in the main text, rather than citing them only in the response letter. To merit publication in a high-impact journal, the authors are encouraged to clearly address the issues outlined above.

Thank you. All references we cited in our previous response letter were cited in the main text except for one, which was an oversight. We now include it as new ref. 42:

42 Hostetler, T., Chen, T.-Y. & Chang, W. Bioinformatic, structural, and biochemical analysis leads to the discovery of novel isonitrilases and decodes their substrate selectivity. *RSC Chem. Biol.* **6**, 583–589 (2025). <https://doi.org/10.1039/D4CB00304G>

Reviewer #2 (Remarks to the Author):

All of my prior concerns have been adequately addressed. One minor point to fix is in the legend to Figure S5; the word "density" is crossed out but should be deleted.

We fixed this typo. Thank you for catching it.

Reviewer #3 (Remarks to the Author):
